# PopSGD: Decentralized Stochastic Gradient Descent in the Population Model

## Abstract

The population model is a standard way to represent large-scale decentralized distributed systems, in which agents with limited computational power interact in randomly chosen pairs, in order to collectively solve global computational tasks. In contrast with synchronous gossip models, nodes are anonymous, lack a common notion of time, and have no control over their scheduling. In this paper, we examine whether large-scale distributed optimization can be performed in this extremely restrictive setting.

We introduce and analyze a natural decentralized variant of stochastic gradient descent (SGD), called PopSGD, in which every node maintains a local parameter, and is able to compute stochastic gradients with respect to this parameter. Every pair-wise node interaction performs a stochastic gradient step at each agent, followed by averaging of the two models. We prove that, under standard assumptions, SGD can converge even in this extremely loose, decentralized setting, for both convex and non-convex objectives. Moreover, surprisingly, in the former case, the algorithm can achieve *linear* speedup in the number of nodes $n$. Our analysis leverages a new technical connection between decentralized SGD and randomized load-balancing, which enables us to tightly bound the concentration of node parameters. We validate our analysis through experiments, showing that PopSGD can achieve convergence and speedup for large-scale distributed learning tasks in a supercomputing environment.

## 1 Introduction

Distributed machine learning has become commonplace, and it is not unusual to encounter systems which distribute model training among tens or even hundreds of nodes. In this paper, we take this trend to the extreme, and ask: would it be possible to distribute basic optimization procedures such as stochastic gradient descent (SGD) to *thousands* of agents? How could the dynamics be implemented in such a large-scale setting, and what would be with the resulting convergence and speedup behavior?

To get some intuition, let us consider the classical *data-parallel* distribution strategy for SGD Bottou (2010). We are in the classical empirical risk minimization setting, where we have a set of samples $S$ from a distribution, and wish to minimize the function $f : \mathbb{R}^d \to \mathbb{R}$, which is the average of losses over samples from $S$ by finding $x^\star = \text{argmin}_x \sum_{s \in S} f_s(x)/|S|$. Assume that we have $P$ compute nodes which can process samples in parallel. Data-parallel SGD consists of parallel iterations, in which each node computes the gradient for one sample, followed by a gradient exchange. Globally, this leads to the iteration:

$$x_{t+1} = x_t - \eta_t \sum_{i=1}^{P} \widetilde{g}_t^i(x_t),$$

where $\eta_t$ is the learning rate, $x_t$ is the value of the global parameter, initially $0^d$, and $\widetilde{g}_t^i(x_t)$ is the stochastic gradient with respect to the parameter obtained by node $i$ at time $t$.

When extending this strategy to high node counts, two major bottlenecks are *communication* and *synchronization*. In particular, to maintain a consistent view of the parameter $x_t$, the nodes would need to broadcast and receive all gradients, and would need to synchronize with all other nodes, at the end of every iteration. Recently, a tremendous amount of work has been dedicated to address these two barriers. In particular, there has been significant progress on *communication-reduced* variants of SGD (e.g. Seide et al. (2014); Strom (2015); Alistarh et al. (2017b); Wen et al. (2017); Aji and Heafield (2017); Dryden et al. (2016); Grubic et al. (2018)), *asynchronous* variants (e.g. Recht et al. (2011); Sa et al. (2015); Duchi et al. (2015); Alistarh et al. (2018b)), as well as *large-batch* or *periodic model averaging* methods, which aim to reduce the *frequency* of communication(e.g. Goyal et al. (2017); You et al. (2017) and Chen and Huo (2016); Stich (2018)), or even *decentralized synchronous*

variants(e.g. Lian et al. (2017a); Tang et al. (2018); Koloskova et al. (2019)). Using such techniques, it is possible to scale SGD to hundreds of nodes, even for complex objectives such as the training of deep neural networks. However, in systems with node counts in the thousands or larger, some of the communication and synchronization requirements of these algorithms become infeasible.

In this paper, we consider the classic *population* model of distributed computing Angluin et al. (2006), defined as follows. We are given a population of $n$ compute agents, each with its own input, which cooperate to perform some globally meaningful computation with respect to their inputs. Interactions occur *pairwise*, where the two interaction partners are *randomly chosen* in every step. Thus, algorithms are specified in terms of the agents' state transitions upon an interaction. The basic unit of time is a single pairwise interaction between two nodes, whereas global (parallel) time is measured as the total number of interactions divided by $n$, the number of nodes. Parallel time corresponds intuitively to the *average* number of interactions per node to reach convergence. Population protocols have a rich history in distributed computing (e.g. Angluin et al. (2006; 2007; 2008a;b;c); Alistarh et al. (2017a; 2018a)), and are standard in modelling distributed systems with millions or billions of nodes, such as Chemical Reaction Networks (CRNs) Bower and Bolouri (2004); Chen et al. (2017) and synthetic DNA strand displacement cascades Chen et al. (2013). The key difference between population protocols and the synchronous gossip models (e.g. Xiao and Boyd (2004); Lian et al. (2017a); Koloskova et al. (2019)) previously used to analyze decentralized SGD is that nodes are *not synchronized*: since interactions occur randomly at arbitrary times, there are no global rounds, and nodes lack a common notion of time.

While the population model is a theoretical construct, we show that it can be efficiently mapped to large-scale super-computing scenarios, with large numbers of compute nodes connected by a fast point-to-point interconnect, where we can avoid the high costs of global synchronization.

An immediate instantiation of SGD in the population model would be to initially assign one sample $s^i$ from the distribution to each node $i$, and have each node maintain its own parameter estimate $x^i$. Whenever two nodes interact, they exchange samples, and each performs a gradient update with respect to the other's sample. If we assume interaction pairs are *uniform random* (with replacement), each node would obtain a stochastic gradient upon each interaction, and therefore each model would converge locally. However, this instance would not have any parallel speedup, since the SGD instances at each node are essentially independent.

In this context, a natural change to the above procedure is to have nodes $i$ and $j$ first perform a gradient step, and then also *average* their resulting models upon every interaction. Effectively, if node $i$ interacts with node $j$, node $i$'s updated model becomes

$$x^i \leftarrow \frac{x^i + x^j}{2} - \eta^i \frac{\widetilde{g}^i(x^i) + \widetilde{g}^j(x^j)}{2}, \tag{1.1}$$

where $j$ is the interaction partner, and the stochastic gradients $\widetilde{g}^i$ and $\widetilde{g}^j$ are taken with respect to each other's samples. The update for node $j$ is symmetric. In this paper, we analyze a variant of the above protocol, which we call PopSGD, in the population protocol model.

We show that, perhaps surprisingly, this simple decentralized SGD averaging dynamic provides strong convergence guarantees for both convex and non-convex objectives. First, we prove that, under standard convexity and smoothness assumptions, PopSGD has convergence speedup that *linear* in the number of nodes $n$. Second, we show that, if the objective is non-convex but satisfies the Polyak-Łojasiewicz (PL) assumption, PopSGD can *still* ensure linear convergence in the number of nodes. Third, we show a $\Theta(\sqrt{n})$ speedup in the non-convex case in the absence of this assumption, matching or slightly improving results from previous work which considered similar models (Lian et al., 2017b).

On the practical side, we provide convergence and speedup results using an efficient implementation of PopSGD using Pytorch/MPI applied to regression tasks, but also to the standard CIFAR/ImageNet classification tasks for deployments on a multi-GPU nodes, and on the Piz Daint supercomputer (Piz). Experiments confirm the scalability of PopSGD. We also observe an improvement in *convergence* versus number of SGD iterations per model at higher node counts, in both convex and non-convex settings. In particular, using PopSGD, we are able to train the ResNet18 and ResNet50 He et al. (2016) models to full accuracy using only $1/8$ the number of SGD updates per model, compared to the sequential baseline, resulting in fast convergence with nearly linear scalability.

This suggests that, even though interactions occur only pairwise, uniformly at random, and in an uncoordinated manner, as long as the convergence time is large enough to amortize the information propagation, the protocol enjoys the full parallel speedup of mini-batch SGD with a batch size proportional to the number of nodes. While similar speedup behaviour has been observed in various *synchronous* models for the convex case– e.g. Stich (2018); Koloskova et al. (2019), or for complex *accelerated* algorithms Hendrikx et al. (2018)–we are the first to show that SGD does not require the existence of globally synchronized rounds or global communication.

Central to our analytic approach is a new technical connection between averaging decentralized SGD and the line of research studying *load-balancing* processes in computer science (e.g. Azar et al. (1999); Mitzenmacher (2000); Talwar and Wieder (2007); Peres et al. (2015a); Boyd et al. (2006)). Intuitively, we show PopSGD can be viewed as a composition between a set of instances of SGD–each corresponding to one of the local parameters $x^i$–which are loosely coupled via pairwise averaging, whose role is to "balance" the models by keeping them well concentrated around their mean, despite the random nature of the pairwise interactions. Our analysis characterizes this concentration, showing that, in essence, the averaging process propagates enough information to globally "simulate" SGD with a batch of size $\Theta(n)$, even though communication is only performed pairwise. We emphasize that the convexity of the objective function in isolation would not be sufficient to prove this fact: simply averaging $n$ independent SGD models at the end of training would not lead to speedup in the objective (please see e.g. Stich (2018) for a detailed discussion). Along the way, we overcome non-trivial technical difficulties, such as the lack of a common notion of time among nodes, or the fact that, due to the structure of SGD, this novel load-balancing process exhibits non-trivial correlations within the same round.

**Related Work.** The study of decentralized optimization algorithms dates back to Tsitsiklis (1984), and is related to the study of *gossip* algorithms for information dissemination Kempe et al. (2003); Xiao and Boyd (2004); Boyd et al. (2006). Gossip is usually studied in one of two models Boyd et al. (2006): *synchronous*, structured in global rounds, where each node interacts with a randomly chosen neighbor, and *asynchronous*, where each node wakes up at times given by a local Poisson clock, and picks a random neighbor to interact with. The population model is functionally equivalent to the asynchronous gossip model, since the interaction times in the latter model can be "discretized" to lead to pairwise uniform interactions. The key difference between our work and averaging in the gossip model, e.g. Boyd et al. (2006), is that their input model is *static* (node inputs are fixed, and node estimates must converge to the true mean), whereas we study the a *dynamic* setting, where the models are updated in each round by SGD, and should remain concentrated around the parameter mean as it converges towards the optimum. Several optimization algorithms have been analyzed in this setting Nedic and Ozdaglar (2009); Johansson et al. (2009); Shamir and Srebro (2014). Tang et al. (2018); Koloskova et al. (2019) analyze quantization in the synchronous gossip model.

Lian et al. (2017a;b); Assran et al. (2018) consider SGD-type algorithms in the gossip model. Specifically, they analyze the same SGD averaging dynamic in the non-convex setting. Table 2 in the appendix summarizes their assumptions, results, and rates. Their results are phrased in the synchronous gossip model, in which nodes interact in a sequence of perfect matchings, for which they provide $O(1/\sqrt{Tn})$ convergence rates (under analytical assumptions). Further, they extend their results to a variant of the gossip model where updates can be performed based on stale information.

Upon careful examination, we find that their results can be extended to the population protocol/asynchronous gossip model, although at the cost of slower convergence relative to the synchronous case. For Lian et al. (2017b), the convergence rate bound requires that the total number of iterations is $\Omega(n^6)$, while in our case only $O(n^4)$ iterations are needed. The difference comes from the fact that our bound on the potential $\Gamma$ is tighter. For Assran et al. (2018), speedup with respect to the number of nodes depends on the parameter $C$. Which in turn, depends on the dimension of the objective function, number iterations for the graph given by edge sets of all matrices used in averaging to be connected and the diameter of aforementioned graph. Unfortunately, in the population model parameter $C$ will not be a constant and this will eliminate the speedup. Further, these references present scalability results for training neural networks using both synchronous and asynchronous variants of their algorithms, thereby slightly relaxing their analytic assumptions.

Relative to this prior work, our contributions are as follows. First, we consider both convex and non-convex objectives. We are the first to show *linear* speedup in the objective for convex objectives in this decentralized asynchronous model. Our bounds in the non-convex case match or slightly improve those presented above, under similar assumptions. Furthermore, under the PL condition,

we are the first to show *linear* convergence speedup in the non-convex case. Our analysis technique relies on a fine-grained analysis of individual interactions, which is different than that of previous work. From the implementation perspective, the performance of our algorithm is competitive with that of previous methods, notably DA-PSGD Lian et al. (2017b) and SGP Assran et al. (2018).

Other instances in the literature which consider dynamic interaction models are Nedic et al. Nedic et al. (2017), who present a gradient tracking algorithm in a different dynamic graph model, and Hendrickx et al. Hendrikx et al. (2018), who achieve exponential convergence rates in a gossip model where transmissions are synchronized across *edges*. The algorithm they consider is a more complex instance of accelerated coordinate descent, and is therefore quite different from the simple dynamics we consider. Neither reference considers large-scale deployments for non-convex objectives.

## 2 PRELIMINARIES

**The Population Protocol Model.** We consider a variant of the population protocol model which consists of a set of $n \geq 2$ anonymous agents, or nodes, each executing a local state machine. (Our analysis will make use of node identifiers only for exposition purposes.) Since our application is continuous optimization, we will assume that the agents' states may store real numbers. The execution proceeds in discrete *steps*, where in each step a new pair of agents is selected uniformly at random to interact from the set of all possible pairs. (To preserve symmetry of the protocols, we will assume that a process may interact with a copy of itself, with low probability.) Each of the two chosen agents updates its state according to a state update function, specified by the algorithm. The basic unit of time is a single pairwise interaction between two nodes. Notice however that in a real system $\Theta(n)$ of these interactions could occur in parallel. Thus, a standard global measure is *parallel time*, defined as the total number of interactions divided by $n$, the number of nodes. Parallel time intuitively corresponds to the *average* number of interactions per node to convergence.

**Stochastic Optimization.** We assume that the agents wish to minimize a $d$-dimensional, differentiable and strongly convex function $f : \mathbb{R}^d \to \mathbb{R}$ with parameter $\ell > 0$, that is:

$$(x - y)^T (\nabla f(x) - \nabla f(y)) \geq \ell \|x - y\|^2, \forall x, y \in \mathbb{R}^d. \tag{2.1}$$

Specifically, we will assume the empirical risk minimization setting, in which agents are given access to a set of data samples $S = \{s_1, \ldots, s_m\}$ coming from some underlying distribution $\mathcal{D}$, to a function $f_i : \mathbb{R}^d \to \mathbb{R}$ which encodes the loss of the argument at the sample $s_i$. The goal of the agents is to converge on a model $x^*$ which minimizes the empirical loss, that is

$$x^* = \operatorname{argmin}_x f(x) = \operatorname{argmin}_x (1/m) \sum_{i=1}^{m} f_i(x). \tag{2.2}$$

In this paper, we assume that the agents employ these samples to run a decentralized variant of SGD, described in detail in the next section. For this, we will assume that agents have access to *stochastic gradients* $\widetilde{g}$ of the function $f$, which are functions such that $\mathbb{E}[\widetilde{g}(x)] = \nabla f(x)$. Stochastic gradients can be computed by each agent by sampling i.i.d. the distribution $D$, and computing the gradient of $f$ at $\theta$ with respect to that sample. In the population model, we could implement this by procedure either by allowing agents to sample in each step, or by assigning a sample $s_i$ to each agent $i$, and having agents compute gradients of their local models with respect to each others' samples. We will assume the following about the gradients:

- **Smooth Gradients**: The gradient $\nabla f(x)$ is $L$-Lipschitz continuous for some $L > 0$, i.e. for all $x, y \in \mathbb{R}^d$:

$$\|\nabla f(x) - \nabla f(y)\| \leq L\|x - y\|. \tag{2.3}$$

- **Bounded Variance**: The variance of the stochastic gradients is bounded by some $\sigma^2 > 0$, i.e. for all $x \in \mathbb{R}^d$:

$$\mathbb{E}\left\|\widetilde{g}(x) - \mathbb{E}[\widetilde{g}(x)]\right\|^2 \leq \sigma^2. \tag{2.4}$$

- **Bounded Second Moment**: The second moment of the stochastic gradients is bounded by some $M^2 > 0$, i.e. for all $x \in \mathbb{R}^d$:

$$\mathbb{E}\|\widetilde{g}(x)\|^2 \leq M^2. \tag{2.5}$$

## 3 THE POPULATION SGD ALGORITHM

**Algorithm Description.** We now describe a decentralized variant of SGD, designed to be executed by a population of $n$ nodes, interacting in uniform random pairs as per the population protocol model. We assume that each node $i$ has access to local stochastic gradients $\widetilde{g}^i$, and maintains a model estimate $X^i$, as well as a local learning rate $\eta^i$. For simplicity, we will assume that this initial estimate is $0^d$ at each agent, although its value may be arbitrary. We detail the way in which the learning rates are updated below. Specifically, upon every interaction, the interacting agents $i$ and $j$ perform the following steps:

```
1  % i and j are chosen uniformly at random, with replacement
2  upon each interaction between agents i and j
3      % each agent performs a local SGD step
4      X^i ← X^i − η^i g̃^i(X^i)
5      X^j ← X^j − η^j g̃^j(X^j)
6      % agents average their estimates coordinate-wise
7      avg ← (X^i + X^j)/2
8      X^i ← avg
9      X^j ← avg
```

**Algorithm 1:** Population SGD pseudocode for each interaction between arbitrary nodes $i$ and $j$.

We are interested in the convergence of local models : $X^1, X^2, ..., X^n$ after $T$ interactions occur in total. For the theoretical reasons, in the case when $f$ is convex, we derive convergence for $y_T$ which is weighted average of average values of local models per step(See Theorem 4.1). In the beginning of section 5 we show that by performing single global averaging step at time step $0 \leq t < T$, which is carefully chosen from specified distribution, we can make sure that in expectation local models converge with the same rate as $y_T$.

**Estimating Time and the Learning Rate.** In parallel with the above algorithm, each agent maintains a local time value $V^i$, which is estimated using a local "phase clock" protocol. These local times are defined and updated as follows. The initial value at each agent is $V^i = 0$. Upon each interaction, the interacting agents $i$ and $j$ exchange their time values. The agent with a *lower* time value, say $V^i < V^j$, will increment its value by 1(ties are broken arbitrarily). The other agent keeps its local value unchanged. (We break ties arbitrarily.) Although intuitively simple, the above procedure provides strong probabilistic guarantees on how far individual values may stray from the mean: with high probability,[1] all the estimates $V^i$ are in the interval $[t/n - c\log T, t/n + c\log T]$, where $c$ is a constant.

Given the current value of $V^i$ at the agent, the value of the learning rate at $i$ is simply $\eta^i = b/(nV^i+a)$, where $a$ and $b$ are constant parameters which we will fix later. This will ensure that the gap between two agents' learning rates will be in the interval $[0.5, 2]$, w.h.p. (See Lemma 4.2.)

## 4 THE CONVERGENCE OF POPSGD IN THE CONVEX CASE

This section is dedicated to proving that the following result holds with high probability:

**Theorem 4.1.** *Let $f$ be an $L$-smooth, $\ell$-strongly convex function satisfying conditions (2.3)—(2.5), whose minimum $x^\star$ we are trying to find via the PopSGD procedure given in Algorithm 1. Let the learning rate for process $i$ at local time $t^i = nV_t^i$ be $\eta_t^i = b/(t^i + a)$, where $a = \max(2cn\log T, 18n, 256L/\ell)$ and $b = 4n/\ell$ are fixed(for some constant $c$). Let the sequence of weights $w_t$ be given by $w_t = (a + t)^2$. Define $\mu_t = \sum_{i=1}^n X_t^i$, $S_T = \sum_{t=0}^{T-1} w_t \geq \frac{1}{3}T^3$ and $y_T = \frac{1}{S_T}\sum_{t=0}^{T-1} w_t\mu_t$. Then, for any time $T$, we have with probability $1 - O(1/\operatorname{poly} T)$ that*

$$\mathbb{E}[f(y_T) - f(x^*)] \leq \frac{a^3\ell}{2S_T}\|\mu_0 - x^*\|^2 + \frac{64T(T + 2a)}{\ell S_T}\sigma^2 + \frac{9216Tn^2}{\ell^2 S_T}M^2 L. \tag{4.1}$$

**Discussion.** We first emphasize that, in the above bound, the time $T$ refers to the number of interactions (as opposed to *parallel time*). With this in mind, we focus on the bound in the case where $T \gg n$, and the parameters $M$, $L$, and $\ell$ are assumed to be well-behaved. In this case, since $S_T \geq T^3/3$, the first and third terms are vanishing as $T$ grows, and we get that convergence is

---

[1]An event holds *with high probability (w.h.p.)* if it occurs with probability $\geq 1 - 1/T^\gamma$, for constant $\gamma > 0$ and the total number of interactions - $T$.

dominated by the second term, which can be bounded as $O(\sigma^2/T)$. It is tempting to think that this is roughly the same rate as *sequential* SGD; however, our notion of time is *different*, as we are counting the total number of SGD steps executed in total *at all the models*. (In fact, the total number of SGD steps up to $T$ is $2T$, since each interaction performs two SGD steps.)

It is interesting to interpret this from the perspective of an arbitrary local model. For this, notice that the *parallel time* corresponding to the number of total interactions $T$, which is by definition $T_p = T/n$, corresponds (up to constants) to the *average* number of interactions and SGD steps performed by each node up to time $T$. Thus, for any single model, convergence with respect to its number of performed SGD steps $T_p$ would be $O(\sigma^2/(nT_p))$, which would correspond to running SGD with a batch size of $n$. Notice that this reduction in convergence time is solely thanks to the *averaging* step: in the absence of averaging, each local model would converge independently at a rate of $O(\sigma^2/T_p)$. We note that our discussion assumes a batch size of 1, but it would generalize to arbitrary batch size $b$, replacing $\sigma^2$ with $\sigma^2/b$. We note that, due to the concentration properties of the averaging process, the claim above can be extended to show convergence behavior for arbitrary individual models (instead of the average of models $\mu_T$).

**Proof Overview.** The argument, given in full in the Additional Material, can be split into two steps. The first step aims to bound the variance of the local models $X_t^i$ at each time $t$ and node $i$ with respect to the mean $\mu_t = \sum_i X_t^i/n$. It views this quantity as a potential $\Gamma_t$, which we show has supermartingale-like behavior, which enables us to bound its expected value as $O(\eta_t^2 n)$. This shows that the variance of the parameters is always bounded with respect to the number of nodes, but also, importantly, that it can be controlled via the learning rate. The key technical step here is Lemma 4.3, which provides a careful bound for the evolution of the potential at a step, by modelling SGD as a dynamic load balancing process: each interaction corresponds to a *weight generation* step (in which gradients are generated) and a *load balancing step*, in which the "loads" of the two nodes (corresponding to their model values) are balanced through averaging.

In the second step of the proof, we first bound the rate at which the mean $\mu_t$ converges towards $x^*$, where we crucially (and carefully) leverage the variance bound obtained above. This is our second key technical lemma. Next, with this in hand, we can apply a standard argument to characterize the rate at which the quantity $\mathbb{E}[f(y_T) - f(x^*)]$ converges towards 0.

**Notation and Preliminaries.** In this section, we overview the analysis of the PopSGD protocol. We begin with some notation. Recall that $n$ is the number of nodes. We will analyze a sequence of *time steps* $t = 1, 2, \ldots, T$, each corresponding to an individual interaction between two nodes, which are usually denoted by $i$ and $j$. Recall the definition of *parallel time* $T_p = T/n$, where $T$ counts the number of pairwise interactions. For any time $t$, define by $\eta_t = b/(a + t)$ the "true" learning rate at time $t$, where $a$ and $b$ are constants to be fixed later, such that $a \geq 2cn \log n$ for some constant $c$. We denote by $x^*$ the optimum of the function $f$.

**Learning Rate Estimates.** Our first technical result characterizes the gap between the "global" learning rate $\eta_t = b/(a + t)$ (in terms of the true time $t$), and the individual learning rates at an arbitrary agent $i$ at the same time, denoted by $\eta_t^i$.

**Lemma 4.2.** *Let $\eta_t^i = b/(a + nV_t^i)$, be the learning rate estimate of agent $i$ at time step $t$, in terms of its time estimate $V_t^i$. Then, there exists a constant $\gamma > 1$ such that, with probability at least $1 - 1/T^\gamma$ (Here, $T$ is a total number of steps our algorithms takes), the following holds for every $T \geq t \geq 0$ and agent $i$:*

$$\frac{1}{2} \leq \frac{\eta_t}{\eta_t^i} \leq 2. \tag{4.2}$$

**Step 1: Parameter Concentration.** Next, let $X_t$ be a vector of model estimates at time step $t$, that is $X_t = (X_t^1, X_t^2, \ldots, X_t^n)$. Also, let $\mu_t = \frac{1}{n} \sum_{i=1}^{n} X_t^i$, be an average estimate at time step $t$. The following potential function measures the concentration of the models around the average:

$$\Gamma_t = \sum_{i=1}^{n} \|X_t^i - \mu_t\|^2.$$

With this in place, one of our key technical results is to provide a supermartingale-type bound on the evolution of the potential $\Gamma_t$, in terms of $M$, $\eta_t$, and the number of nodes $n$.

**Lemma 4.3.** *For any time step $t$ and fixed learning rate $\eta_t$ used at $t$, we have the bound*

$$\mathbb{E}[\Gamma_{t+1}|\Gamma_t] \leq \left(1 - \frac{1}{n}\right)\Gamma_t + 4\eta_t M\left(\frac{\Gamma_t}{n}\right)^{1/2} + 8\eta_t^2 M^2.$$

Next, we unroll this recurrence to upper bound $\Gamma_t$ in expectation for any time step $t$, by choosing an appropriate series of non-constant learning rates.

**Lemma 4.4.** *If $a \geq 18n$, then the potential is bounded as follows*

$$\mathbb{E}[\Gamma_t] \leq 36nb^2/(t+a)^2 M^2 = 36n\eta_t^2 M^2.$$

**Step 2: Convergence of the Mean and Risk Bound.** The above result allows us to characterize how well the individual parameters are concentrated around their mean, in terms of the second moment of the gradients, the number of nodes, and the learning rate. In turn, this will allow us to provide a recurrence for how fast the parameter average is moving towards the optimum, in terms of the variance and second-moment bounds of the gradients:

**Lemma 4.5.** *For $\eta_t \leq \frac{n}{64L}$, we have that*

$$\mathbb{E}\left\|\mu_{t+1} - x^*\right\|^2 \leq \left(1 - \frac{\eta_t \ell}{n}\right)\mathbb{E}\|\mu_t - x^*\|^2 - \frac{\eta_t}{2n}\mathbb{E}[f(\mu_t) - f(x^*)] + \frac{16\sigma^2\eta_t^2}{n^2} + \frac{288\eta_t^3 M^2 L}{n}.$$

Finally, we wish to phrase this bound as a recurrence which will allow us to bound the expected risk of the weighted sum average. We aim to use the following standard result (see e.g. Stich (2018)):

**Lemma 4.6.** *Let $\{a_t\}_{t\geq 0}$, $a_t \geq 0$, $\{e_t\}_{t\geq 0}$, $e_t \geq 0$ be sequences satisfying*

$$a_{t+1} \leq \left(1 - \ell\alpha_t\right)a_t - \alpha_t e_t A + \alpha_t^2 B + \alpha_t^3 C,$$

*for $\alpha_t = \frac{4}{\ell(t+a)}$, $A > 0$, $B, C \geq 0$, $\ell > 0$ then*

$$\frac{A}{S_T}\sum_{t=0}^{T-1} w_t e_t \leq \frac{\ell a^3}{4S_T}a_0 + \frac{2T(T+2a)}{\ell S_T}B + \frac{16T}{\ell^2 S_T}C, \tag{4.3}$$

*for $w_t = (a+t)^2$ and $S_T = \sum_{t=0}^{T-1} w_T \geq \frac{1}{3}T^3$.*

To use the above lemma, we set $\eta_t = n\alpha_t = \frac{4n}{\ell(t+a)}$, and the parameter $b = 4n/\ell$. We also use $A = 1/2$, $B = 16\sigma^2$, and $C = 288M^2Ln^2$. Let $y_T = \frac{1}{nS_T}\sum_{i=1}^{n}\sum_{t=0}^{T-1} w_t X_t^i$. Also, let $e_t = \mathbb{E}[f(\mu_t) - f(x^*)]$ and $a_t = \mathbb{E}\left\|\mu_t - x^*\right\|^2$.

Using convexity and Lemma 4.6 above we obtain the following final bound:

$$\mathbb{E}[f(y_T) - f(x^*)] \leq \frac{a^3\ell}{2S_T}\|\mu^0 - x^*\|^2 + \frac{64T(T+2a)}{\ell S_T}\sigma^2 + \frac{9216Tn^2}{\ell^2 S_T}M^2 L. \tag{4.4}$$

To complete the proof of the Theorem, we only need to find the appropriate value of the parameter $a$. For that, we list all the constraints on $a$: $a \geq 2cn\log T$, $a \geq 18n$ and $\frac{4n}{\ell(t+a)} \leq \frac{n}{64L}$. These inequalities can be satisfied by setting $a = \max\left(2cn\log T, 18n, 256\frac{L}{\ell}\right)$. This concludes our proof.

## 5 EXTENSIONS

**Convergence of local models and alternative to computing $y_T$.** Notice that Theorem 4.1 measures convergence of $f(y_T)$, where $y_T = \sum_{t=0}^{T-1} \frac{w_t}{S_T}\frac{\sum_{i=1}^{n} X_t^i}{n} = \sum_{t=0}^{T-1} \frac{w_t}{S_T}\mu_t$, is a weighted average of $\mu_t$-s per step. Notice that actually computing $y_T$ can be expensive, since we need values of local models over $T$ steps and it does not necessarily guarantee convergence of each individual model. In order to circumvent this issue, we can look at the following inequality, which in combination with the Jensen's inequality gives us the proof of Theorem 4.1 (Please see Appendix for details) :

$$\frac{1}{S_T}\sum_{t=0}^{T-1} w_t\mathbb{E}[f(\mu_t) - f(x^*)] \leq \frac{a^3\ell}{2S_T}\|\mu_0 - x^*\|^2 + \frac{64T(T+2a)}{\ell S_T}\sigma^2 + \frac{9216Tn^2}{\ell^2 S_T}M^2 L. \tag{5.1}$$

What we can do is, instead of computing $y_T$, we just sample time step $0 \leq t \leq T-1$ with probability $\frac{w_t}{S_T}$ and compute $f(\mu_t) = f(\sum_{i=1}^{n} X_t^i/n)$, by using single global averaging procedure. Observe that $\mathbb{E}_t[\mathbb{E}_{\mu_t}[f(\mu_t)]]$ is exactly the left hand side of the above inequality.

Hence, we get the convergence identical to the one in Theorem 4.1 and additionally, since we are using global averaging, we also guarantee the same convergence for each local model. Finally, we would like to emphasize that in practice there is no need to compute $y_T$ or to use global averaging, since local models are already converged after $T$ interactions.

**General Interaction Graphs.** Our analysis can be extended to more general interaction graphs by tying the evolution of the potential in this case. In the following, we present the results for a *cycle*, leaving the exact derivations for more general classes of expander graphs for the full version. In particular, we assume that each agent is a node on a cycle, and that it is allowed to interact only with its neighbouring nodes. Again, the scheduler chooses interaction edges uniformly at random. In this setting, we can show the following result, which is similar to Theorem 4.1:

**Theorem 5.1.** *Let $f$ be an $L$-smooth, $\ell$-strongly convex function satisfying conditions (2.3)—(2.5), whose minimum $x^\star$ we are trying to find via the PopSGD procedure on a cycle. Let the learning rate for process $i$ at local time $t^i = nV_t^i$ be $\eta_t^i = b/(t^i+a)$, where $a = \max(2cn\log T, 18n, 256L/\ell)$ and $b = 4n/\ell$ are fixed(for some constant c). Let the sequence of weights $w_t$ be given by $w_t = (a+t)^2$. Define $\mu_t = \sum_{i=1}^n X_t^i$, $S_T = \sum_{t=0}^{T-1} w_t \geq \frac{1}{3}T^3$ and $y_T = \frac{1}{S_T}\sum_{t=0}^{T-1} w_t\mu_t$. Then, for any time $T$, we have with probability $1 - O(1/\operatorname{poly} T)$ that*

$$E[f(y_T) - f(x^*)] \leq \frac{a^3\ell}{2S_T}\|\mu^0 - x^*\|^2 + \frac{64T(T+2a)}{\ell S_T}\sigma^2 + \frac{25600Tn^6}{\ell^3 S_T}M^2L^2.$$

Notice that for $T \gg n^3$, the second term dominates convergence and we can repeat the same argument as for Theorem 4.1 to show $O(\sigma^2/T)$ convergence (where $T$ is the total number of interactions). Next we provide the sketch of a proof for the PopSGD on a cycle case. See section **??** in the appendix for the proof sketch.

**The Non-Convex Case.** Next, we show convergence for non-convex, but smooth functions. The Following theorem deals with the bounded gradient case:

**Theorem 5.2.** *Let $f$ be an non-convex, $L$-smooth, function satisfying assumption 2.5, whose minimum $x^\star$ we are trying to find via the PopSGD procedure given in Algorithm 1. Let the learning rate we use be $\eta = n/\sqrt{T}$. Then, for any $T \geq n^4$:*

$$\frac{1}{T}\sum_{t=0}^{T-1} \mathbb{E}\|\nabla f(\mu_t)\|^2 \leq \frac{(f(\mu_0) - f(x^*))}{\sqrt{T}} + \frac{36LM^2}{\sqrt{T}} + \frac{2LM^2}{\sqrt{T}}.$$

Next, we show the similar result for the case when gradient is not bounded, but it's variance is. This can be achieved by carefully following steps given in the Lian et al. (2017b).

**Theorem 5.3.** *Let $f$ be an non-convex, $L$-smooth, function satisfying condition 2.4, whose minimum $x^\star$ we are trying to find via the PopSGD procedure given in Algorithm 1. Let the learning rate we use be $\eta = n/\sqrt{T}$. Then, for any $T \geq 4624 \max\{1/L^2, 1\}n^4$, we have*

$$\sum_{t=0}^{T-1} \frac{1}{T}\mathbb{E}\|f(\mu_t)\|^2 \leq \frac{\mathbb{E}[f(\mu_0)] - \mathbb{E}[f(x^*)]}{\sqrt{T}} + \frac{4\sigma^2}{\sqrt{T}} + \frac{32L^2\sigma^2}{\sqrt{T}} + \frac{768L^2\sigma^2}{\sqrt{T}}.$$

Observe that, since $T$ is the total number of interactions and is equal to $nT_p$, where $T_p$ is a parallel time, in both theorems, we get convergence $O(1/\sqrt{T}) = O(1/\sqrt{T_p n})$. Which gives us $1/\sqrt{n}$ speedup over $O(1/\sqrt{T})$ convergence of the sequential version. (Note that in the sequential case parallel time and the total number of interactions are the same.)

Finally, we derive convergence for the case where gradient is bounded and function we are trying to optimize satisfies the Polyak-Łojasiewicz (PL) assumption with constant $\alpha$. More formally, function $f$ satisfies the PL assumption with constant $\alpha$ if :

$$\frac{1}{2}\mathbb{E}\|\nabla f(\mu_t)\|^2 \geq \alpha(f(\mu_t) - f(x^*)). \tag{5.2}$$

We use a similar approach to Haddadpour et al. (2019), but, our tighter analysis of $\Gamma$ potential allows us to remove global synchronization.

**Theorem 5.4.** *Let $f$ be an non-convex, $L$-smooth, function satisfying assumption 2.5 and $PL$-assumption with constant $\alpha$(5.2), whose minimum $x^\star$ we are trying to find via the PopSGD procedure*

given in Algorithm 1. Let the learning rate at time step $t$ be $\eta_t = \frac{4n}{\alpha(t+a)}$ (Local learning rates at time $T_i$ are $\eta_t^i = \frac{4n}{\alpha(T^i+a)}$). Then for $a \geq 2cn \log T$ and time $T$, with probability $1 - O(1/\operatorname{poly} T)$ we have

$$\mathbb{E}[f(\mu_T)] - f(x^*) \leq \frac{a^3}{(a+T)^3}(\mathbb{E}[f(\mu_0)] - f(x^*)) + \frac{4608L^2M^2n^2T}{\alpha^3(T+a)^3} + \frac{64LM^2}{\alpha^2(T+a)}.$$

Observe that as in Theorem 4.1, this gives us a linear speedup over the sequential version.

## 6 EXPERIMENTAL RESULTS

In this section, we validate our results numerically by implementing PopSGD in Pytorch, using MPI for inter-node communication MPI. We are interested in the *convergence* behavior of the algorithm, and in the *scalability* with respect to the number of nodes. Our study is split into simulated experiments for convex objectives–to examine the validity of our analysis as $n$ increases—and large-scale real-world experiments for non-convex objectives (training neural networks), aimed to examine whether PopSGD can provide scalability and convergence for such objectives.

**Convex Objectives.** To validate our analysis in the convex case, we evaluated the performance of PopSGD on three datasets: (1) a real-world linear regression problem (the *Year Prediction* dataset Chang and Lin (2011)) with a $463, 715/51, 630$ test/train split, and $d = 90$; (2) a real-world classification problem (*gisette* Chang and Lin (2011)) with $6, 000/1, 000$ test/train split, and $d = 5000$; (3) a synthetic least-squares problem of the form (2.2) with $f(x) = \frac{1}{2}\|Ax - b\|^2$, where $A \in \mathbb{R}^{m \times d}$ and $b \in \mathbb{R}^m$, with $m = 10^4$ and variable $d$. As a baseline, we employ vanilla SGD with manual learning rate tuning. The learning rate is adjusted in terms of the number of local steps each node has taken, similar to our analysis.

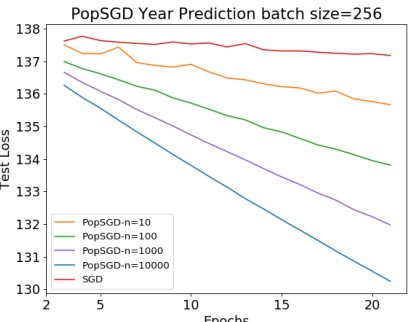
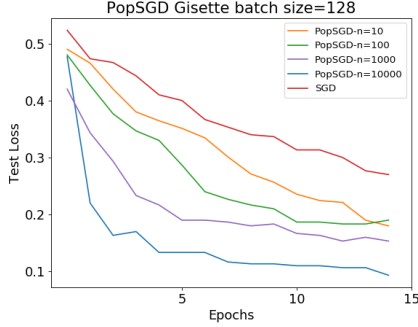

(a) PopSGD test loss vs. $n$ on Year Prediction.  (b) PopSGD test loss vs. $n$ on Gisette.

Figure 1: PopSGD convergence (test loss at the step versus parallel time) for various node counts $n$ on a real linear regression (left) and logistic regression (right) datasets. The baseline is sequential SGD, which is identical to PopSGD with node count 1.

Our first set of experiments examines train and test loss for PopSGD on the real-world tasks specified above. We examine the *test loss* behavior with respect to the number of nodes $n$, and execute for powers of 10 between 1 and 10000. Each node obtains a stochastic gradient by sampling 128 elements from the training set in a batch. We tuned the learning rate parameter for each instance independently, through line search, and obtained learning rates in the interval $[0.0005, 0.015]$ for Gisette, and $[0.05, 0.2]$ for Year Prediction.

Please see Figure 1(b) for the results.(The number of epochs is cropped to maintain visibility, but the trends are maintained in general.) The results confirm our analysis; notice in particular the clear separation between instances for different $n$, which follows exactly the increase in the number of nodes, although the X axis values correspond to the same number of gradient steps for the local model. In Appendix B, we present additional experiments which precisely examine the reduction in variance versus the number of nodes on the synthetic regression task, confirming our analysis.

**Training Neural Networks.** Our second set of experiments tests PopSGD on the CSCS Piz Daint supercomputer, which is composed of Cray XC50 nodes, each with a Xeon E5-2690v3 CPU and an NVIDIA Tesla P100 GPU, using a state-of-the-art Aries interconnect. For this, we implemented

PopSGD in Pytorch using MPI one-sided primitives MPI, which allow nodes to read eachothers' models for averaging without explicit synchronization. We used PopSGD to train ResNets on the classic CIFAR-10 and ImageNet datasets.

Training proceeds in epochs, each of which is structured as follows. At the beginning of each epoch, we shuffle the dataset and partition it among processes. Notice that, in data-parallel SGD, an epoch ends after each process iterates exactly once over its partition, i.e. each sample is seen *once*. However, Theorem 4.1 suggests that, for PopSGD, processes should iterate several times over their partitions, for the corresponding gradient information to be propagated. To match this, we introduce a multiplier constant mult, which counts the number of times each process will iterate through its partition before an epoch is complete. At the same time, we *scale down* the total number of epochs executed by $n$, the number of nodes. In practical terms, if sequential SGD trains ResNet50 in 90 epochs, decreasing the learning rate at 30 and 60 epochs, then PopSGD with 32 nodes and multiplier 4 would use $90 * 4/32 \simeq 12$ epochs per node, decreasing the learning rate at 4 and 8 epochs.

Since PopSGD scales almost linearly in terms of time per epoch (see Figure 2, middle), this should ensure end-to-end speedup for PopSGD. In particular, for ResNet50, we obtain a 2x end-to-end time-to-convergence speedup versus data-parallel SGD. Figure 2 shows the test and train accuracies for the ResNet18 model trained on the ImageNet dataset, with 32 Piz Daint nodes and mult $= 4$, as well as scalability versus number of nodes. The hyperparameters used for model training are identical to the standard sequential recipe (batch size 128 per node), with the number of epochs scaled down to 12 per node.

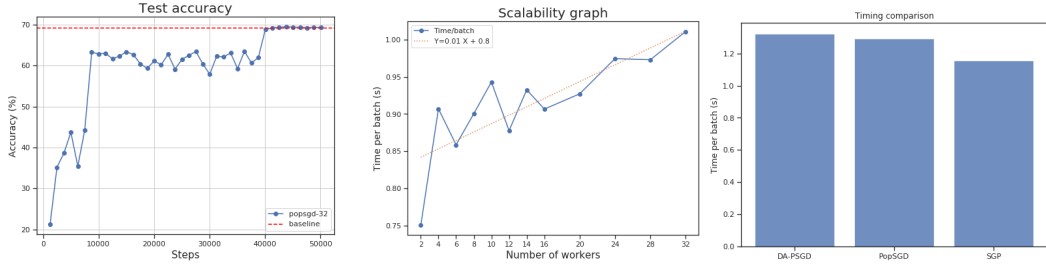

Figure 2: PopSGD test accuracy using 32 nodes on Piz Daint, measured at a fixed arbitrary node. The X axis measures SGD steps per model, whereas the $Y$ axis measures Top-1 accuracy. The dotted red line is the accuracy of the Torchvision baseline. PopSGD surpasses the test accuracy of the baseline by $0.34\%$, although it processes each sample $4\times$ less times, and each model sees $8\times$ less gradient updates. The third graph shows the average runtime per batch for PopSGD (center) versus DA-PSGD (Lian et al., 2017b) and SGP (Assran et al., 2018) on the same setup.

The results suggest that PopSGD can indeed preserve convergence, while being scalable and competitive with state-of-the-art algorithms. Appendix B presents additional experiments for ResNet50/Imagenet and ResNet20/CIFAR-10, which further substantiate this claim.

## 7 DISCUSSION AND FUTURE WORK

We have analyzed the convergence of decentralized SGD in the population model of distributed computing. We have shown that SGD is able to still converge in this restrictive setting, and moreover, under parameter and objective assumptions, can even achieve *linear* speedup in the number of agents $n$ in terms of parallel time. The empirical results confirmed our analytical findings. The main surprising result is that PopSGD presents speedup behavior roughly similar to mini-batch SGD, even though a node only sees one gradient update and a single model at a time. Our work opens several avenues for extensions. One natural direction is to study PopSGD with quantized communication, or allowing the interactions to present inconsistent (stale) model views to the two agents. Another avenue is to tighten the bounds in terms of their dependence on the problem conditioning, and on the objective assumptions.

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

## A  SUMMARY AND COMPARISON OF RESULTS

In this section we compare convergence rates of existing algorithms, while specifying the bounds they require for convergence. In the tables $T$-corresponds to the parallel time and $n$ is a number of processes. We use the following notations for needed bounds:

1. $\sigma^2$ - bound on the variance of gradient.
2. $M^2$ - bound on the second moment of gradient.
3. **PL** - Polyak-Łojasiewicz assumption.
4. **d** - bounded dimension.
5. $\rho$ - bounded spectral gap of the averaging matrix.
6. $\tau$ - bounded message delay.

|  | Global synchronization | Assumptions | Convergance Rate |
|---|---|---|---|
| PopSGD | NO | $\sigma^2, M^2$ | $O(1/Tn)$ |
| Local SGD | YES | $\sigma^2, M^2$ | $O(1/Tn)$ |

Table 1: **convex case**

|  | Global synchronization | Assumptions | Convergance Rate |
|---|---|---|---|
| PopSGD | NO | $\sigma^2, M^2$ | $O(1/\sqrt{Tn})$ |
| PopSGD | NO | $\sigma^2, M^2, PL$ | $O(1/Tn)$ |
| PopSGD | NO | $\sigma^2$ | $O(1/\sqrt{Tn})$ |
| LUPA-SGD | YES | $\sigma^2, M^2, PL$ | $O(1/Tn)$ |
| AD-SGD | NO | $\sigma^2, \rho, \tau$ | $O(1/\sqrt{Tn})$ |
| SGP | NO | $\sigma^2, d, \tau$ | $O(1/\sqrt{Tn})$ |

Table 2: **non-convex case**

## B  ADDITIONAL EXPERIMENTS

**Convex Losses.** In these experiments, we examine the convergence of PopSGD versus parallel time for different node counts, and compared it with the sequential baseline. More precisely, for PopSGD, we execute the protocol by simulating the entire sequence of interactions sequentially, and track the evolution of train and test loss at an arbitrary fixed model $x^i$ with respect to the number of SGD steps it performs. Notice that this is practically equivalent to tracking with respect to *parallel* time. In this case, the theory suggests that loss convergence and variance should both improve when increasing the number of nodes. Figure 3(a) presents the results for the synthetic linear regression example with $d = 32$, for various values of $n$, for constant learning rate $\eta = 0.001$ across all models, and batch size 1 for each local gradient. Figure 3(b) compares PopSGD convergence (with local batch size 1) against sequential mini-batch SGD with batch size equal to the number of nodes $n$.

Examining Figure 3(a), we observe that both the convergence and loss variance improve as we increase the number of nodes $n$, even though the target model executes exactly the same number of gradient steps at the same point on the $x$ axis. Of note, variance decreases proportionally with the number of nodes, with $n = 128$ having the smallest variance. Compared to mini-batch SGD with batch size $= n$ (Figure 3(b)), PopSGD with $n = 128$ has similar, but notably higher variance, which follows the analytical bound in Theorem 4.1.

**CIFAR-10 Experiments.** We illustrate convergence and scaling results for non-convex objectives by using PopSGD to train a standard ResNet20 DNN model on CIFAR-10 in Pytorch, using 8 GPU nodes, comparing against vanilla and local SGD performing global averaging every 100 batches (we found this value necessary for the model to converge). We measure the error/loss at an arbitrary process for PopSGD. We run the parallel versions at 4 and 8 nodes.

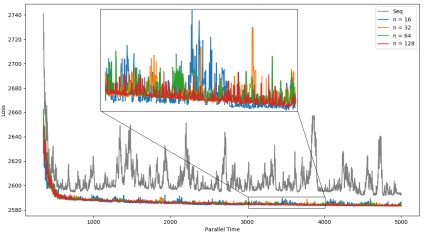

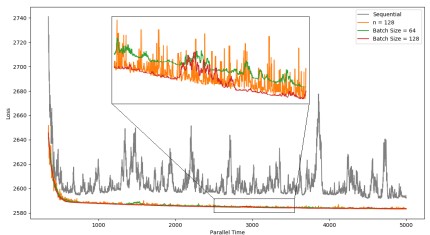

(a) PopSGD convergence vs. $n$.

(b) PopSGD convergence vs. mini-batch SGD.

Figure 3: PopSGD convergence (training loss at the step versus parallel time) on the synthetic regression task versus the number of nodes $n$ (left), and versus sequential SGD with different batch sizes (right). Sequential SGD is identical to PopSGD with node count 1. The cutouts represent zoomed views.

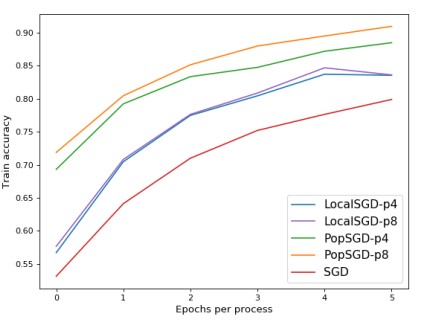

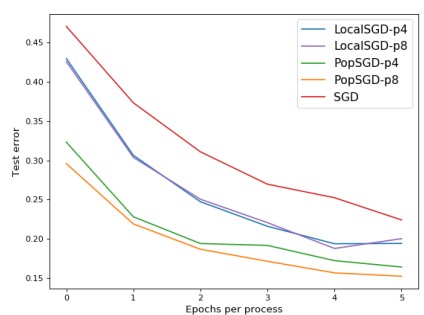

(a) Train accuracy for ResNet20/CIFAR10.

(b) Test error for ResNet20/CIFAR10.

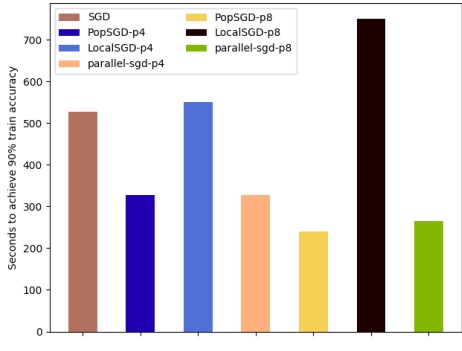

(c) Time to 90% train accuracy.

The results in Figure 4(c) show that (a,b) PopSGD does indeed converge faster as we increase population size, tracking the trend from the convex case; and (c) PopSGD can provide non-trivial scalability, comparable or better than data-parallel and local SGD.

**Training ResNet50 on ImageNet.** Figure 4 shows the test and train accuracies for the ResNet50 model trained on the ImageNet dataset, with 32 Piz Daint nodes and $\mathsf{mult} = 4$. PopSGD achieves test accuracy within $< 0.5\%$ relative to the Torchvision baseline, despite the vastly inferior number of iterations, in a total of 29 hours. By way of comparison, end-to-end training using standard data-parallel SGD takes approximately 48h on the same setup.

## C  COMPLETE CORRECTNESS ARGUMENT

**Lemma 4.2.** *Let $\eta_t^i = b/(a + nV_t^i)$, be the learning rate estimate of agent $i$ at time step $t$, in terms of its time estimate $V_t^i$. Then, there exists a constant $\gamma > 1$ such that, with probability at least $1 - 1/T^\gamma$*

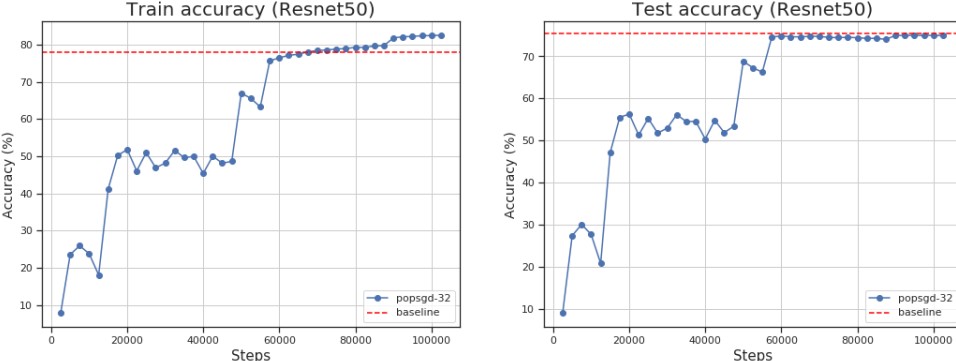

Figure 4: PopSGD train and test accuracy using 32 nodes on Piz Daint, measured at a fixed arbitrary node, for training ResNet50 on ImageNet. The round multipler value is $\mathsf{mult} = 4$. The X axis measures SGD steps per model, whereas the $Y$ axis measures Top-1 accuracy. The dotted red line is the accuracy of the Torchvision baseline (Marcel and Rodriguez, 2010). PopSGD is below the test accuracy of the baseline by $< 0.5\%$.

*(Here, $T$ is a total number of steps our algorithms takes), the following holds for every $T \geq t \geq 0$ and agent $i$:*

$$\frac{1}{2} \leq \frac{\eta_t}{\eta_t^i} \leq 2. \tag{C.1}$$

*Proof.* Let $G^t = \sum_{i=1}^{n} \exp\left(\zeta(V_t^i - \frac{t}{n})\right) + \sum_{i=1}^{n} \exp\left(-\zeta(V_t^i - \frac{t}{n})\right)$, for some fixed constant $\zeta$. The following lemma is proved as Theorem 2.10 in Peres et al. (2015b):

**Lemma C.1.** *For any $t \geq 0$, and some fixed constants $\epsilon$ and $\theta$, $\mathbb{E}[G_t] \leq \frac{4\theta}{\zeta\epsilon}n$.*

Subsequently, we can show that for any $t \geq 0$ and agent $i$:

$$Pr\left[ \mid \frac{t}{n} - V_t^i \mid \geq \frac{q}{\zeta} \log T \right] \leq Pr[G^t \geq T^q] \overset{Markov}{\leq} \frac{4\theta}{\zeta\epsilon} \frac{n}{T^q}. \tag{C.2}$$

Hence, for large enough constant $q$, using union bound over $T$ steps, we can show that there exists a constant $\gamma > 0$ such that for every $T \geq t \geq 0$ and agent $i$, $\mid \frac{t}{n} - V_t^i \mid \leq \frac{q}{\zeta} \log T$, with probability at least $1 - 1/T^\gamma$.

Let $c$ be $\frac{q}{\zeta}$, thus $a \geq 2cn \log T = 2\frac{q}{\zeta}n \log T$. This allows us to finish the proof of the lemma:

$$\frac{1}{2} \leq \frac{a + t - \frac{q}{\zeta}n \log T}{a + t} \leq \frac{\eta_t}{\eta_t^i} \leq \frac{a + t + \frac{q}{\zeta}n \log T}{a + t} \leq 2. \tag{C.3}$$

$\square$

This allows us to bound the per step change of potential $\Gamma$, in terms of global learning rate $\eta_t$.

**Lemma 4.3.** *For any time step $t$ and fixed learning rate $\eta_t$ used at $t$, we have the bound*

$$\mathbb{E}[\Gamma_{t+1}|\Gamma_t] \leq \left(1 - \frac{1}{n}\right)\Gamma_t + 4\eta_t M\left(\frac{\Gamma_t}{n}\right)^{1/2} + 8\eta_t^2 M^2.$$

*Proof.* First we bound change in potential $\Delta_t = \Gamma_{t+1} - \Gamma_t$ for some time step $t > 0$. Let $\Delta_t^{i,j}$ be a change in potential when we choose different agents $i$ and $j$ at random and let $\Delta_t^i$ be a change in potential when we select the same node $i$. We get that

$$\mathbb{E}\left[\Delta_t|X_t\right] = \sum_{i} \sum_{i \neq j} \frac{1}{n^2}\mathbb{E}\left[\Delta_t^{i,j}|X_t\right] + \sum_{i=1}^{n} \frac{1}{n^2}\mathbb{E}\left[\Delta_t^i|X_t\right]. \tag{C.4}$$

We proceed by bounding a change in potential for fixed $i \neq j$. Observe, that in this case $\mu_{t+1} = \mu_t - (\eta_t^i \widetilde{g}_i(X_t^i) + \eta_t^j \widetilde{g}_j(X_t^j))/n$ and $X_{t+1}^i = X_{t+1}^j = (X_t^i + X_t^j)/2 - (\eta_t^i \widetilde{g}_i(X_t^i) + \eta_t^j \widetilde{g}_j(X_t^j))/2$.

Hence,

$$X_{t+1}^i - \mu_{t+1} = X_{t+1}^j - \mu_{t+1} = (X_t^i + X_t^j)/2 - \frac{n-2}{2n}(\eta_t^i \widetilde{g}_i(X_t^i) + \eta_t^j \widetilde{g}_j(X_t^j)) - \mu_t.$$

For $k \notin \{i, j\}$, since $X_{t+1}^k = X_t^k$ we get that

$$X_{t+1}^k - \mu_{t+1} = X_t^k + \frac{1}{n}(\eta_t^i \widetilde{g}_i(X_t^i) + \eta_t^j \widetilde{g}_j(X_t^j)) - \mu_t.$$

This gives us that

$$
\begin{aligned}
\mathbb{E}\left[\Delta_t^{i,j}|X_t\right] = \quad & \mathbb{E}\left\|(X_t^i + X_t^j)/2 - \frac{n-2}{2n}(\eta_t^i \widetilde{g}_i(X_t^i) + \eta_t^j \widetilde{g}_j(X_t^j)) - \mu_t\right\|^2 - \|X_t^i - \mu_t\|^2 \\
& + \mathbb{E}\left\|(X_t^i + X_t^j)/2 - \frac{n-2}{2n}(\eta_t^i \widetilde{g}_i(X_t^i) + \eta_t^j \widetilde{g}_j(X_t^j)) - \mu_t\right\|^2 - \|X_t^j - \mu_t\|^2 \\
& + \sum_{k \notin \{i,j\}}\left(\mathbb{E}\left\|X_t^k + \frac{1}{n}(\eta_t^i \widetilde{g}_i(X_t^i) + \eta_t^j \widetilde{g}_j(X_t^j)) - \mu_t\right\|^2 - \|X_t^k - \mu_t\|^2\right) \\
= \quad & 2\|(X_t^i - \mu_t)/2 + (X_t^j - \mu_t)/2\|^2 - \|X_t^i - \mu_t\|^2 - \|X_t^j - \mu_t\|^2 \\
& - \frac{n-2}{n}\mathbb{E}\langle \eta_t^i \widetilde{g}_i(X_t^i) + \eta_t^j \widetilde{g}_j(X_t^j), (X_t^i - \mu_t) + (X_t^j - \mu_t)\rangle \\
& + 2\left(\frac{n-2}{2n}\right)^2 \mathbb{E}\|\eta_t^i \widetilde{g}_i(X_t^i) + \eta_t^j \widetilde{g}_j(X_t^j)\|^2 \\
& + \sum_{k \notin \{i,j\}}\left(\frac{2}{n}\mathbb{E}\langle \eta_t^i \widetilde{g}_i(X_t^i) + \eta_t^j \widetilde{g}_j(X_t^j), X_t^k - \mu_t\rangle + \frac{1}{n^2}\mathbb{E}\|\eta_t^i \widetilde{g}_i(X_t^i) + \eta_t^j \widetilde{g}_j(X_t^j)\|^2\right)
\end{aligned}
$$

Observe that

$$\mathbb{E}\|\eta_t^i \widetilde{g}_i(X_t^i) + \eta_t^j \widetilde{g}_j(X_t^j)\|^2 \leq 2(\eta_t^i)^2 \mathbb{E}\|\widetilde{g}_i(X_t^i)\|^2 + 2(\eta_t^j)^2 E\|\widetilde{g}_j(X_t^j)\|^2 \overset{\text{Fact (2.5)}}{\leq} 2M^2\left((\eta_t^i)^2 + (\eta_t^j)^2\right) \overset{\text{Lemma 4.2}}{\leq} 16\eta_t^2 M^2.$$

and

$$\sum_{k=1}^n \mathbb{E}\langle \eta_t^i \widetilde{g}_i(X_t^i) + \eta_t^j \widetilde{g}_j(X_t^j), X_t^k - \mu_t\rangle = 0.$$

Thus, we have that

$$
\begin{aligned}
\mathbb{E}\left[\Delta_t^{i,j}|X_t\right] \leq \quad & 2\|(X_t^i - \mu_t)/2 + (X_t^j - \mu_t)/2\|^2 - \|X_t^i - \mu_t\|^2 - \|X_t^j - \mu_t\|^2 \\
& - \mathbb{E}\langle \eta_t^i \widetilde{g}_i(X_t^i) + \eta_t^j \widetilde{g}_j(X_t^j), (X_t^i - \mu_t) + (X_t^j - \mu_t)\rangle \\
& + 32\eta_t^2\left(\frac{n-2}{2n}\right)^2 M^2 + \sum_{k \notin \{i,j\}}\frac{16}{n^2}\eta_t^2 M^2 \\
\leq \quad & -\|X_t^i - \mu_t\|^2/2 - \|X_t^j - \mu_t\|^2/2 + \langle X_t^i - \mu_t, X_t^j - \mu_t\rangle \\
& - \mathbb{E}\langle \eta_t^i \widetilde{g}_i(X_t^i) + \eta_t^j \widetilde{g}_j(X_t^j), (X_t^i - \mu_t) + (X_t^j - \mu_t)\rangle \\
& + 8\eta_t^2 M^2. \quad\quad\quad\quad (C.6)
\end{aligned}
$$

similarly we can prove that

$$\mathbb{E}\left[\Delta_t^i|X_t\right] \leq -\mathbb{E}\langle \eta_t^i \widetilde{g}_i(X_t^i) + \eta_t^i \widetilde{g}_i(X_t^i), (X_t^i - \mu_t) + (X_t^i - \mu_t)\rangle + 8\eta_t^2 M^2. \quad\quad (C.7)$$

By using inequalities C.6 and C.7 in inequality C.4 we get that

$$
\begin{aligned}
\mathbb{E}\left[\Delta_t|X_t\right] = \quad & \sum_i \sum_{i \neq j}\frac{1}{n^2}\mathbb{E}\left[\Delta_t^{i,j}|X_t\right] + \sum_{i=1}^n \frac{1}{n^2}\mathbb{E}\left[\Delta_t^t|X_t\right] \\
\leq \quad & -\sum_i \sum_{i \neq j}\frac{1}{n^2}\left(\|X_t^i - \mu_t\|^2/2 + \|X_t^j - \mu_t\|^2/2\right) + \sum_i \sum_{i \neq j}\frac{1}{n^2}\langle X_t^i - \mu_t, X_t^j - \mu_t\rangle
\end{aligned}
$$

$$-\sum_i \sum_j \frac{1}{n^2} \mathbb{E}\langle \eta_t^i \widetilde{g}_i(X_t^i) + \eta_t^j \widetilde{g}_j(X_t^j), (X_t^i - \mu_t) + (X_t^j - \mu_t)\rangle + 8\eta_t^2 M^2.$$

Observe that

$$\sum_i \sum_{i \neq j} \frac{1}{n^2}\langle X_t^i - \mu_t, X_t^j - \mu_t\rangle = \sum_{i=1}^n \frac{1}{n^2}\langle X_t^i - \mu_t, \sum_{j \neq i} X_t^j - \mu_t\rangle = \frac{1}{n^2}\sum_i -\|X_t^i - \mu_t\|^2 = -\frac{1}{n^2}\Gamma_t.$$

and

$$\sum_i \sum_{i \neq j} \frac{1}{n^2}\Big(\|X_t^i - \mu_t\|^2/2 + \|X_t^j - \mu_t\|^2/2\Big) = \frac{n-1}{n^2}\sum_i \|X_t^i - \mu_t\|^2 = \frac{n-1}{n^2}\Gamma_t.$$

Hence, we get that

$$\mathbb{E}\Big[\Delta_t | X_t\Big] \leq -\frac{\Gamma_t}{n} - \sum_i \sum_j \frac{1}{n^2}\mathbb{E}\langle \eta_t^i \widetilde{g}_i(X_t^i) + \eta_t^j \widetilde{g}_j(X_t^j), (X_t^i - \mu_t) + (X_t^j - \mu_t)\rangle + 8\eta_t^2 M^2. \quad \text{(C.9)}$$

Further, we have that

$$\sum_i \sum_j \frac{1}{n^2}\mathbb{E}\langle \eta_t^i \widetilde{g}_i(X_t^i) + \eta_t^j \widetilde{g}_j(X_t^j), (X_t^i - \mu_t) + (X_t^j - \mu_t)\rangle$$

$$= \sum_i \sum_j \frac{1}{n^2}\mathbb{E}\langle \eta_t^i \widetilde{g}_i(X_t^i), (X_t^j - \mu_t)\rangle + \sum_i \sum_j \frac{1}{n^2}\mathbb{E}\langle \eta_t^j \widetilde{g}_j(X_t^j), (X_t^i - \mu_t)\rangle + \sum_{i=1}^n \frac{2\eta_t^i}{n}\mathbb{E}\langle \widetilde{g}_i(X_t^i), X_t^i - \mu_t\rangle$$

$$= \sum_{i=1}^n \frac{2\eta_t^i}{n}\mathbb{E}\langle \widetilde{g}_i(X_t^i), X_t^i - \mu_t\rangle \overset{\text{Cauchy-Schwarz}}{\leq} \sum_{i=1}^n \frac{2\eta_t^i}{n}\mathbb{E}\Big[\|\widetilde{g}_i(X_t^i)\| \cdot \|X_t^i - \mu_t\|\Big]$$

$$= \sum_{i=1}^n \frac{2\eta_t^i}{n}\mathbb{E}\Big[\|\widetilde{g}_i(X_t^i)\|\Big]\|X_t^i - \mu_t\| \overset{\text{Jensen}}{\leq} \sum_{i=1}^n \frac{2\eta_t^i}{n}\Big(\mathbb{E}\|\widetilde{g}_i(X_t^i)\|^2\Big)^{\frac{1}{2}}\|X_t^i - \mu_t\| \leq \sum_{i=1}^n \frac{2\eta_t^i M}{n}\|X_t^i - \mu_t\|$$

$$\overset{\text{Lemma 4.2}}{\leq} \frac{4\eta_t M}{n}\sum_{i=1}^n \|X_t^i - \mu_t\| \overset{\text{Cauchy-Schwarz}}{\leq} \frac{4\eta_t M}{n}\Big(n\sum_{i=1}^n \|X_t^i - \mu_t\|^2\Big)^{1/2} = 4\eta_t M\Big(\frac{\Gamma_t}{n}\Big)^{1/2}.$$

By plugging above inequality in inequality C.9, we get that

$$\mathbb{E}[\Delta_t | X_t] \leq -\frac{\Gamma_t}{n} + 4\eta_t M\Big(\frac{\Gamma_t}{n}\Big)^{1/2} + 8\eta_t^2 M^2.$$

Hence, considering the definition of $\Delta_t$ and the fact that the above inequality implies

$$\mathbb{E}[\Delta_t | \Gamma_t] \leq -\frac{1}{n}\Gamma_t + 4\eta_t M\Big(\frac{\Gamma_t}{n}\Big)^{1/2} + 8\eta_t^2 M^2,$$

we get the proof of the Lemma. $\qquad\square$

**Lemma 4.4.** *If $a \geq 18n$, then the potential is bounded as follows*
$$\mathbb{E}[\Gamma_t] \leq 36nb^2/(t+a)^2 M^2 = 36n\eta_t^2 M^2.$$

*Proof.* We prove the lemma using induction. Base case $t = 0$ trivially holds, since $\Gamma_t = 0$. For induction step, we assume that at time step $t$, $\mathbb{E}[\Gamma_t] \leq 36nb^2 M^2/(t+a)^2$. Our goal is to prove that $\mathbb{E}[\Gamma_{t+1}] \leq 36nb^2 M^2/(t+a+1)^2$.

$$\mathbb{E}[\Gamma_{t+1}] = \mathbb{E}[\mathbb{E}[\Gamma_{t+1}|\Gamma_t]] \overset{\text{Lemma 4.3}}{\leq} \Big(1 - \frac{1}{n}\Big)\mathbb{E}[\Gamma_t] + 4\eta_t M\mathbb{E}\Big[\Big(\frac{\Gamma_t}{n}\Big)^{1/2}\Big] + 8\eta_t^2 M^2$$

$$\overset{\text{Jensen}}{\leq} \Big(1 - \frac{1}{n}\Big)\mathbb{E}[\Gamma_t] + 4\eta_t M\Big(\mathbb{E}\Big[\frac{\Gamma_t}{n}\Big]\Big)^{1/2} + 8\eta_t^2 M^2$$

$$\leq \Big(1 - \frac{1}{n}\Big)\frac{36nb^2 M^2}{(t+a)^2} + \frac{24b^2 M^2}{(t+a)^2} + \frac{8b^2 M^2}{(t+a)^2}$$

$$\leq \frac{36nb^2 M^2}{(t+a+1)^2} + \Big(\frac{36nb^2 M^2}{(t+a)^2} - \frac{36nb^2 M^2}{(t+a+1)^2}\Big) - \frac{4b^2 M^2}{(t+a)^2}$$

$$= \frac{36b^2M^2}{(t+a+1)^2} + \frac{36nb^2M^2\big(2(t+a)+1\big)}{(t+a)^2(t+a+1)^2} - \frac{4b^2M^2}{(t+a)^2}$$

$$\leq \frac{36nb^2M^2}{(t+a+1)^2} + \frac{72nb^2M^2(t+a+1)}{(t+a)^2(t+a+1)^2} - \frac{4b^2M^2}{(t+a)^2}.$$

Using the fact that $t + a + 1 \geq a \geq 18n$ in the above inequality allows us to get

$$\mathbb{E}[\Gamma_{t+1}] \leq \frac{36nb^2M^2}{(t+a+1)^2} + \frac{4b^2M^2}{(t+a)^2} - \frac{4b^2M^2}{(t+a)^2} \leq \frac{36nb^2M^2}{(t+a+1)^2}.$$

$\square$

**Lemma 4.5.** *For $\eta_t \leq \frac{n}{64L}$, we have that*

$$\mathbb{E}\left\|\mu_{t+1} - x^*\right\|^2 \leq \left(1 - \frac{\eta_t\ell}{n}\right)\mathbb{E}\|\mu_t - x^*\|^2 - \frac{\eta_t}{2n}\mathbb{E}[f(\mu_t) - f(x^*)] + \frac{16\sigma^2\eta_t^2}{n^2} + \frac{288\eta_t^3M^2L}{n}.$$

*Proof.* Let $F_t$ be the amount by which $\mu_t$ decreases at step $t$. So, $F_t$ is a sum of $\frac{\eta_t^i}{n}\widetilde{g}_i(X_t^i)$ and $\frac{\eta_t^j}{n}\widetilde{g}(X_t^j)$ for agents $i$ and $j$, which interact at step $t$. Also, let $F_t'$ be the amount by which $\mu_t$ would decrease if interacting agents used true gradients. That is, for agents $i$ and $j$ which interact at step $t$, $F_t'$ is sum of $\frac{\eta_t^j}{n}\nabla f(X_t^i)$ and $\frac{\eta_t^j}{n}\nabla f(X_t^j)$.

$$\mathbb{E}\left\|\mu_{t+1} - x^*\right\|^2 = \mathbb{E}\left\|\mu_t - F_t - x^*\right\|^2 = \mathbb{E}\left\|\mu_t - F_t - x^* - F_t' + F_t'\right\|^2$$

$$= \mathbb{E}\left\|\mu_t - x^* - F_t'\right\|^2 + \mathbb{E}\left\|F_t' - F_t\right\|^2 + 2\mathbb{E}\left\langle\mu_t - x^* - F_t', F_t' - F_t\right\rangle \quad \text{(C.10)}$$

Observe that $\mathbb{E}[F_t] = F_t'$, hence the last term in the equation above is $0$. This means that in order to upper bound $\mathbb{E}\left\|\mu_{t+1} - x^*\right\|^2$, we need to upper bound $\mathbb{E}\left\|\mu_t - x^* - F_t'\right\|^2$ and $\mathbb{E}\left\|F_t' - F_t\right\|^2$.

For the latter, we get that

$$\mathbb{E}\left\|F_t' - F_t\right\|^2 = \frac{1}{n^2}\sum_{i=1}^{n}\sum_{j=1}^{n}\mathbb{E}\left\|\frac{\eta_t^i}{n}(\widetilde{g}_i(X_t^i) - \nabla f(X_t^i)) + \frac{\eta_t^j}{n}(\widetilde{g}_j(X_t^j) - \nabla f(X_t^j))\right\|^2$$

$$\leq \frac{2}{n^2}\sum_{i=1}^{n}\sum_{j=1}^{n}\left(\left(\frac{\eta_t^i}{n}\right)^2\mathbb{E}\|\widetilde{g}_i(X_t^i) - \nabla f(X_t^i)\|^2 + \left(\frac{\eta_t^j}{n}\right)^2\mathbb{E}\|\widetilde{g}_j(X_t^j) - \nabla f(X_t^j)\|^2\right)$$

$$= \frac{4}{n}\sum_{i=1}^{n}\left(\frac{\eta_t^i}{n}\right)^2\mathbb{E}\|\widetilde{g}_i(X_t^i) - \nabla f(X_t^i)\|^2 \overset{\text{Fact (2.4)}}{\leq} \frac{4}{n}\sum_{i=1}^{n}\left(\frac{\eta_t^i}{n}\right)^2\sigma^2 \overset{\text{Lemma 4.2}}{\leq} 16\frac{\sigma^2\eta_t^2}{n^2}.$$

For the former, we have that

$$\mathbb{E}\left\|\mu_t - x^* - F_t'\right\|^2 = \mathbb{E}\|\mu_t - x^*\|^2 + \mathbb{E}\|F_t'\|^2 - 2\mathbb{E}\langle\mu_t - x^*, F_t'\rangle$$

$$= \mathbb{E}\|\mu_t - x^*\|^2 + \frac{1}{n^2}\sum_{i=1}^{n}\sum_{j=1}^{n}\mathbb{E}\|\frac{\eta_t^i}{n}\nabla f(X_t^i) + \frac{\eta_t^j}{n}\nabla f(X_t^j)\|^2$$

$$- \frac{1}{n^2}\sum_{i=1}^{n}\sum_{j=1}^{n}2\mathbb{E}\left\langle\mu_t - x^*, \frac{\eta_t^i}{n}\nabla f(X_t^i) + \frac{\eta_t^j}{n}\nabla f(X_t^j)\right\rangle$$

$$\leq \mathbb{E}\|\mu_t - x^*\|^2 + \frac{4}{n^3}\sum_{i=1}^{n}(\eta_t^i)^2\mathbb{E}\|\nabla f(X_t^i)\|^2 - \frac{4}{n^2}\sum_{i=1}^{n}\mathbb{E}\left\langle\mu_t - x^*, \eta_t^i\nabla f(X_t^i)\right\rangle$$

$$= \mathbb{E}\|\mu_t - x^*\|^2 + \frac{4}{n^3}\sum_{i=1}^{n}(\eta_t^i)^2\mathbb{E}\|\nabla f(X_t^i) - \nabla f(x^*)\|^2 - \frac{4}{n^2}\sum_{i=1}^{n}\mathbb{E}\left\langle\mu_t - X_t^i + X_t^i - x^*, \eta_t^i\nabla f(X_t^i)\right\rangle$$

$$= \mathbb{E}\|\mu_t - x^*\|^2 + \frac{4}{n^3}\sum_{i=1}^n (\eta_t^i)^2 \mathbb{E}\|\nabla f(X_t^i) - \nabla f(x^*)\|^2 - \frac{4}{n^2}\sum_{i=1}^n \eta_t^i \mathbb{E}\Big\langle \mu_t - X_t^i, \nabla f(X_t^i)\Big\rangle$$

$$- \frac{4}{n^2}\sum_{i=1}^n \eta_t^i \mathbb{E}\Big\langle X_t^i - x^*, \nabla f(X_t^i)\Big\rangle \tag{C.11}$$

In order to bound $\|\nabla f(X_t^i) - \nabla f(x^*)\|^2$ we can use the $L-$smoothness property for convex functions, in the following form

$$f(y) \geq f(x) + \langle \nabla f(x), y - x\rangle + \frac{1}{2L}\|\nabla f(y) - \nabla f(x)\|^2. \tag{C.12}$$

By setting $y = X_t^i$ and $x = x^*$ we get

$$\|\nabla f(X_t^i) - \nabla f(x^*)\|^2 \leq 2L(f(X_t^i) - f(x^*)). \tag{C.13}$$

Additionally, by $\ell-$strong convexity, we have that

$$- \Big\langle X_t^i - x^*, \nabla f(X_t^i)\Big\rangle \leq -(f(X_t^i) - f(x^*)) - \frac{\ell}{2}\|X_t^i - x^*\|^2. \tag{C.14}$$

By Cauchy-Schwarz inequality, we get that

$$-2\Big\langle \mu_t - X_t^i, \nabla f(X_t^i)\Big\rangle \leq 2L\|X_t^i - \mu_t\|^2 + \|\nabla f(X_t^i)\|^2/(2L)$$

$$= 2L\|X_t^i - \mu_t\|^2 + \|\nabla f(X_t^i) - \nabla f(x^*)\|^2/(2L)$$

Using $L-$smoothness property (C.13) in the above inequality gives us that

$$-2\Big\langle \mu_t - X_t^i, \nabla f(X_t^i)\Big\rangle \leq 2L\|X_t^i - \mu_t\|^2 + (f(X_t^i) - f(x^*)). \tag{C.15}$$

By plugging inequalities (C.13), (C.14) and (C.15) in inequality (C.11), we get

$$\mathbb{E}\Big\|\mu_t - x^* - F_t'\Big\|^2 \leq \mathbb{E}\|\mu_t - x^*\|^2 + \frac{4L}{n^2}\sum_{i=1}^n \eta_t^i \mathbb{E}\|X_t^i - \mu_t\|^2$$

$$+ \frac{8L}{n^3}\sum_{i=1}^n (\eta_t^i)^2 \mathbb{E}[f(X_t^i) - f(x^*)] - \frac{2}{n^2}\sum_{i=1}^n \eta_t^i \mathbb{E}[f(X_t^i) - f(x^*)]$$

$$- \frac{2\ell}{n^2}\sum_{i=1}^n \eta_t^i \mathbb{E}\|X_t^i - x^*\|^2.$$

Observe that $\mathbb{E}\|X_t^i - x^*\|^2$, $\mathbb{E}\|X_t^i - \mu_t\|^2$ and $\mathbb{E}[f(X_t^i) - f(x^*)]$ are non-negative terms, Thus, by using Lemma 4.2 in the above inequality we have that:

$$\mathbb{E}\Big\|\mu_t - x^* - F_t'\Big\|^2 \leq \mathbb{E}\|\mu_t - x^*\|^2 + \frac{8L\eta_t}{n^2}\sum_{i=1}^n \mathbb{E}\|X_t^i - \mu_t\|^2$$

$$+ \frac{32L\eta_t^2}{n^3}\sum_{i=1}^n \mathbb{E}[f(X_t^i) - f(x^*)] - \frac{\eta_t}{n^2}\sum_{i=1}^n \mathbb{E}[f(X_t^i) - f(x^*)]$$

$$- \frac{\eta_t\ell}{n^2}\sum_{i=1}^n \mathbb{E}\|X_t^i - x^*\|^2$$

$$= \mathbb{E}\|\mu_t - x^*\|^2 + \frac{8L\eta_t}{n^2}\sum_{i=1}^n \mathbb{E}\|X_t^i - \mu_t\|^2$$

$$+ \frac{\eta_t}{n^2}\sum_{i=1}^n \left(\Big(\frac{32L\eta_t}{n} - 1\Big)\mathbb{E}[f(X_t^i) - f(x^*)] - \ell\mathbb{E}\|X_t^i - x^*\|^2\right).$$

By using $\eta_t \leq \frac{n}{64L}$ in the above inequality we get that

$$\mathbb{E}\left\|\mu_t - x^* - F_t'\right\|^2 \leq \mathbb{E}\|\mu_t - x^*\|^2 + \frac{8L\eta_t}{n^2} \sum_{i=1}^{n} \mathbb{E}\|X_t^i - \mu_t\|^2$$

$$+ \frac{\eta_t}{n^2} \sum_{i=1}^{n} \left( -\frac{1}{2}\mathbb{E}[f(X_t^i) - f(x^*)] - \ell\mathbb{E}\|X_t^i - x^*\|^2 \right).$$

By Jensen's inequality and convexity of $f$ and square of norm we have that

$$\mathbb{E}\left\|\mu_t - x^* - \frac{\eta_t}{n}F_t'\right\|^2 \leq \mathbb{E}\|\mu_t - x^*\|^2 + \frac{8\eta_t L}{n^2} \sum_{i=1}^{n} \mathbb{E}\|X_t^i - \mu_t\|^2$$

$$+ \frac{\eta_t}{n} \left( -\frac{1}{2}\mathbb{E}[f(\mu_t) - f(x^*)] - \ell\mathbb{E}\|\mu_t - x^*\|^2 \right)$$

$$= \left(1 - \frac{\eta_t \ell}{n}\right)\mathbb{E}\|\mu_t - x^*\|^2 - \frac{\eta_t}{2n}\mathbb{E}[f(\mu_t) - f(x^*)] + \frac{8\eta_t L}{n^2}\mathbb{E}[\Gamma_t]$$

$$\overset{\text{Lemma 4.4}}{\leq} \left(1 - \frac{\eta_t \ell}{n}\right)\mathbb{E}\|\mu_t - x^*\|^2 - \frac{\eta_t}{2n}\mathbb{E}[f(\mu_t) - f(x^*)]$$

$$+ \frac{288\eta_t^3 M^2 L}{n}$$

Finally, by using the above inequality in inequality (C.10) we get

$$\mathbb{E}\left\|\mu_{t+1} - x^*\right\|^2 \leq \left(1 - \frac{\eta_t \ell}{n}\right)\mathbb{E}\|\mu_t - x^*\|^2 - \frac{\eta_t}{2n}\mathbb{E}[f(\mu_t) - f(x^*)] + \frac{16\sigma^2\eta_t^2}{n^2} + \frac{288\eta_t^3 M^2 L}{n}$$

$$\square$$

**Theorem 4.1.** *Let $f$ be an $L$-smooth, $\ell$-strongly convex function satisfying conditions (2.3)—(2.5), whose minimum $x^*$ we are trying to find via the PopSGD procedure given in Algorithm 1. Let the learning rate for process $i$ at local time $t^i = nV_t^i$ be $\eta_t^i = b/(t^i + a)$, where $a = \max(2cn\log T, 18n, 256L/\ell)$ and $b = 4n/\ell$ are fixed(for some constant $c$). Let the sequence of weights $w_t$ be given by $w_t = (a + t)^2$. Define $\mu_t = \sum_{i=1}^{n} X_t^i$, $S_T = \sum_{t=0}^{T-1} w_t \geq \frac{1}{3}T^3$ and $y_T = \frac{1}{S_T}\sum_{t=0}^{T-1} w_t\mu_t$. Then, for any time $T$, we have with probability $1 - O(1/\text{poly } T)$ that*

$$\mathbb{E}[f(y_T) - f(x^*)] \leq \frac{a^3 \ell}{2S_T}\|\mu_0 - x^*\|^2 + \frac{64T(T + 2a)}{\ell S_T}\sigma^2 + \frac{9216Tn^2}{\ell^2 S_T}M^2 L.$$

*Proof.* We use Lemma 4.6 to solve the recurrence given by Lemma 4.5. For this we set $\eta_t = n\alpha_t = \frac{4n}{\ell(t+a)}$. That is, we set parameter $b = 4n/\ell$. We also use $A = 1/2$, $B = 16\sigma^2$, and $C = 288M^2Ln^2$. This way we can rewrite Lemma 4.5 as :

$$\mathbb{E}\left\|\mu_{t+1} - x^*\right\|^2 \leq (1 - \alpha_t \ell)\mathbb{E}\|\mu_t - x^*\|^2 - A\alpha_t\mathbb{E}[f(\mu_t) - f(x^*)] + B\alpha_t^2 + C\alpha_t^3.$$

Further, let $y_T = \frac{1}{nS_T}\sum_{i=1}^{n}\sum_{t=0}^{T-1} w_t X_t^i$. Also, let $e_t$ be $\mathbb{E}[f(\mu_t) - f(x^*)]$ and $a_t = \mathbb{E}\left\|\mu_t - x^*\right\|^2$.

By convexity of $f$ we have that

$$\mathbb{E}[f(y_T) - f(x^*)] \leq \frac{1}{S_T}\sum_{t=0}^{T-1} w_t\mathbb{E}[f(\mu_t) - f(x^*)] \tag{C.16}$$

Using this fact and the Lemma 4.6 we obtain the following.

$$\mathbb{E}[f(y_T) - f(x^*)] \leq \frac{a^3 \ell}{2S_T}\|\mu^0 - x^*\|^2 + \frac{64T(T + 2a)}{\ell S_T}\sigma^2 + \frac{9216Tn^2}{\ell^2 S_T}M^2 L. \tag{C.17}$$

what is left is to find the appropriate $a$. For that we remember all the constraints on $a$: $a \geq 2cn \log T, a \geq 18n$ and $\frac{4n}{\ell(t+a)} \leq \frac{n}{64L}$. These inequalities can be satisfied by setting $a = \max\left(2cn \log T, 18n, 256\frac{L}{\ell}\right)$.

$\square$

## D  NON CONVEX ANALYSIS WITH $PL$-ASSUMPTION

In this section we deal with the case when the function we are trying to optimize is non convex but it satisfies $PL$-assumption(5.2) with constant $\alpha$. More formally, function $f$ satisfies $PL$-assumption(5.2) with constant $\alpha$.

**Theorem 5.4.** *Let $f$ be an non-convex, $L$-smooth, function satisfying assumption 2.5 and PL-assumption(5.2) with constant $\alpha$, whose minimum $x^\star$ we are trying to find via the PopSGD procedure given in Algorithm 1. Let the learning rate at time step $t$ be $\eta_t = \frac{4n}{\alpha(t+a)}$. Then, for any time $T$, we have*

$$\mathbb{E}[f(\mu_T)] - f(x^*) \leq \frac{a^3}{(a+T)^3}(\mathbb{E}[f(\mu_0)] - f(x^*)) + \frac{4608L^2M^2n^2T}{\alpha^3(T+a)^3} + \frac{64LM^2}{\alpha^2(T+a)}.$$

*Proof.*

$$\mathbb{E}[f(\mu_{t+1})] \overset{L-smoothness}{\leq} \mathbb{E}[f(\mu_t)] + \mathbb{E}\langle \nabla f(\mu_t), \mu_{t+1} - \mu_t \rangle + \frac{L}{2}\mathbb{E}\|\mu_{t+1} - \mu_t\|^2 \tag{D.1}$$

$$= \mathbb{E}[f(\mu_t)] + \sum_{i=1}^{n}\sum_{j=1}^{n}\frac{1}{n^2}\mathbb{E}\langle \nabla f(\mu_t), -\frac{\eta_t^i}{n}\widetilde{g}_i(X_t^i) - \frac{\eta_t^j}{n}\widetilde{g}_j(X_t^j)\rangle \tag{D.2}$$

$$+ \sum_{i=1}^{n}\sum_{j=1}^{n}\frac{L}{2n^2}\mathbb{E}\|\frac{\eta_t^i}{n}\widetilde{g}_i(X_t^i) + \frac{\eta_t^j}{n}\widetilde{g}_j(X_t^j)\|^2 \tag{D.3}$$

$$\leq \mathbb{E}[f(\mu_t)] + \sum_{i=1}^{n}\sum_{j=1}^{n}\frac{1}{n^2}\mathbb{E}\langle \nabla f(\mu_t), -\frac{\eta_t^i}{n}\widetilde{g}_i(X_t^i) - \frac{\eta_t^j}{n}\widetilde{g}_j(X_t^j)\rangle \tag{D.4}$$

$$+ \sum_{i=1}^{n}\sum_{j=1}^{n}\frac{L}{n^2}\mathbb{E}\left[\|\frac{\eta_t^i}{n}\widetilde{g}_i(X_t^i)\|^2 + \|\frac{\eta_t^j}{n}\widetilde{g}_j(X_t^j)\|^2\right] \tag{D.5}$$

$$= \mathbb{E}[f(\mu_t)] + \sum_{i=1}^{n}\frac{2}{n}\mathbb{E}\langle \nabla f(\mu_t), -\frac{\eta_t^i}{n}\widetilde{g}_i(X_t^i)\rangle + \sum_{i=1}^{n}\frac{2L}{n}\mathbb{E}\|\frac{\eta_t^i}{n}\widetilde{g}_i(X_t^i)\|^2. \tag{D.6}$$

Using $\mathbb{E}[\widetilde{g}_i(x)] = \nabla f(x)$ and property (2.5) we can rewrite the above inequality as

$$\mathbb{E}[f(\mu_{t+1})] \leq \mathbb{E}[f(\mu_t)] + \sum_{i=1}^{n}\frac{2}{n}\mathbb{E}\langle \nabla f(\mu_t), -\frac{\eta_t^i}{n}\nabla f(X_t^i)\rangle + \sum_{i=1}^{n}\frac{2L(\eta_t^i)^2}{n^3}M^2 \tag{D.7}$$

$$= \mathbb{E}[f(\mu_t)] + \sum_{i=1}^{n}\frac{2\eta_t^i}{n^2}\mathbb{E}\langle \nabla f(\mu_t), \nabla f(\mu_t) - \nabla f(X_t^i)\rangle \tag{D.8}$$

$$- \sum_{i=1}^{n}\frac{2\eta_t^i}{n^2}\mathbb{E}\|\nabla f(\mu_t)\|^2 + \sum_{i=1}^{n}\frac{2L(\eta_t^i)^2}{n^3}M^2 \tag{D.9}$$

$$\leq \mathbb{E}[f(\mu_t)] + \sum_{i=1}^{n}\frac{\eta_t^i}{n^2}\mathbb{E}\left[\|\nabla f(\mu_t)\|^2 + \|\nabla f(\mu_t) - \nabla f(X_t^i)\|^2\right] \tag{D.10}$$

$$- \sum_{i=1}^{n}\frac{2\eta_t^i}{n^2}\mathbb{E}\|\nabla f(\mu_t)\|^2 + \sum_{i=1}^{n}\frac{2L(\eta_t^i)^2}{n^3}M^2 \tag{D.11}$$

$$= \mathbb{E}[f(\mu_t)] + \sum_{i=1}^{n}\frac{\eta_t^i}{n^2}\mathbb{E}\|\nabla f(\mu_t) - \nabla f(X_t^i)\|^2 - \sum_{i=1}^{n}\frac{\eta_t^i}{n^2}\mathbb{E}\|\nabla f(\mu_t)\|^2 + \sum_{i=1}^{n}\frac{2L(\eta_t^i)^2}{n^3}M^2 \tag{D.12}$$

$$\overset{\text{Lemma 4.2}}{\leq} \mathbb{E}[f(\mu_t)] + \sum_{i=1}^{n} \frac{2\eta_t}{n^2}\mathbb{E}\|\nabla f(\mu_t) - \nabla f(X_t^i)\|^2 - \sum_{i=1}^{n}\frac{\eta_t}{2n^2}\mathbb{E}\|\nabla f(\mu_t)\|^2 + \sum_{i=1}^{n}\frac{8L\eta_t^2}{n^3}M^2$$
(D.13)

$$\overset{L-smoothness}{\leq} \mathbb{E}[f(\mu_t)] + \sum_{i=1}^{n}\frac{2L^2\eta_t}{n^2}\mathbb{E}\|\mu_t - X_t^i\|^2 - \frac{\eta_t}{2n}\mathbb{E}\|\nabla f(\mu_t)\|^2 + \frac{8L\eta_t^2}{n^2}M^2. \quad \text{(D.14)}$$

Recall that by Lemma 4.4 we have that $\mathbb{E}[\Gamma_t] = \sum_{i=1}^{n}\mathbb{E}\|\mu_t - X_t^i\|^2 \leq 36n\eta_t^2 M^2$, hence the above inequality becomes:

$$\mathbb{E}[f(\mu_{t+1})] - \mathbb{E}[f(\mu_t)] \leq \frac{72L^2\eta_t^3 M^2}{n} - \frac{\eta_t}{2n}\mathbb{E}\|\nabla f(\mu_t)\|^2 + \frac{8L\eta_t^2}{n^2}M^2.$$

Next we use $PL$-assumption(5.2), which says that

$$\frac{1}{2}\mathbb{E}\|\nabla f(\mu_t)\|^2 \geq \alpha(f(\mu_t) - f(x*)).$$

Hence, we get that

$$\mathbb{E}[f(\mu_{t+1})] - \mathbb{E}[f(\mu_t)] \leq \frac{72L^2\eta_t^3 M^2}{n} - \frac{\eta_t\alpha}{n}(f(\mu_t) - f(x^*)) + \frac{8L\eta_t^2}{n^2}M^2.$$

This can be rewritten as:

$$\mathbb{E}[f(\mu_{t+1})] - f(x^*) \leq (1 - \frac{\eta_t\alpha}{n})(\mathbb{E}[f(\mu_t)] - f(x^*)) + \frac{72L^2\eta_t^3 M^2}{n} + \frac{8L\eta_t^2}{n^2}M^2. \quad \text{(D.15)}$$

Next as in the proof for convex case, we define $w_t = (a+t)^2$ and we set $\eta_t = \frac{4n}{\alpha(t+a)}$. We get that

$$\frac{w_t}{\eta_t}(1 - \frac{\eta_t\alpha}{n}) = \frac{(a+t)^3\alpha}{4n}(1 - \frac{4}{t+a}) = \frac{(a+t)^2\alpha}{4n}(t+a-4)$$

$$\leq (a+t-1)^3\frac{\alpha}{4n} = \frac{w_{t-1}}{\eta_{t-1}}.$$

By using this in inequality D.15 and unrolling recursion we get that

$$\frac{w_T}{\eta_T}(\mathbb{E}[f(\mu_T)] - f(x^*)) \leq (1 - \frac{\alpha\eta_0}{n})\frac{w_0}{\eta_0}(\mathbb{E}[f(\mu_0)] - f(x^*)) + \sum_{t=0}^{T-1}\frac{w_t}{\eta_t}\frac{72L^2\eta_t^3 M^2}{n} + \sum_{t=0}^{T-1}\frac{w_t}{\eta_t}\frac{8L\eta_t^2}{n^2}M^2$$

$$= (1 - \frac{4}{a})\frac{a^3\alpha}{4n}(\mathbb{E}[f(\mu_0)] - f(x^*)) + \sum_{t=0}^{T-1}\frac{1152L^2 M^2 n}{\alpha^2} + \sum_{t=0}^{T-1}\frac{32(t+a)LM^2}{\alpha n}$$

$$\leq \frac{a^3\alpha}{4n}(\mathbb{E}[f(\mu_0)] - f(x^*)) + \frac{1152L^2 M^2 nT}{\alpha^2} + \frac{16(T+a)^2 LM^2}{\alpha n}.$$

Next we divide the above inequality by $\frac{w_T}{\eta_T} = \frac{(a+T)^3\alpha}{4n}$. We get that

$$\mathbb{E}[f(\mu_T)] - f(x^*) \leq \frac{a^3}{(a+T)^3}(\mathbb{E}[f(\mu_0)] - f(x^*)) + \frac{4608L^2 M^2 n^2 T}{\alpha^3(T+a)^3} + \frac{64LM^2}{\alpha^2(T+a)}.$$

$\square$

## E  NON-CONVEX ANALYSIS WITH CONSTANT LEARNING RATE

In this section we address the case when function we want to optimize is non-convex by using constant learning rate over all iterations and process. Let our learning rate be $eta$. Observe that since $\eta_t^i = \eta_t = \eta \leq 2\eta_t$, Lemma 4.3 holds. So, we get that

$$\mathbb{E}[\Gamma_{t+1}|\Gamma_t] \leq \left(1 - \frac{1}{n}\right)\Gamma_t + 4\eta M\left(\frac{\Gamma_t}{n}\right)^{1/2} + 8\eta^2 M^2. \quad \text{(E.1)}$$

Using induction as in the proof of Lemma 4.4 we can prove that the similar results holds for the constant learning rate as well:

**Lemma E.1.** *For any constant learning rate $\eta$ and $t > 0$, we have*

$$\mathbb{E}[\Gamma_t] \leq 36n\eta^2 M^2. \quad \text{(E.2)}$$

**Theorem 5.2.** *Let $f$ be an non-convex, $L$-smooth, function satisfying assumption 2.5, whose minimum $x^\star$ we are trying to find via the PopSGD procedure given in Algorithm 1. Let the learning*

*rate we use be $\eta = n/\sqrt{T}$. Then, for any $T \geq n^4$:*

$$\frac{1}{T}\sum_{t=0}^{T-1}\mathbb{E}\|\nabla f(\mu_t)\|^2 \leq \frac{(f(\mu_0) - f(x^*))}{\sqrt{T}} + \frac{36LM^2}{\sqrt{T}} + \frac{2LM^2}{\sqrt{T}}.$$

*Proof.*

$$\mathbb{E}[f(\mu_{t+1})] \overset{L-smoothness}{\leq} \mathbb{E}[f(\mu_t)] + \mathbb{E}\langle\nabla f(\mu_t), \mu_{t+1} - \mu_t\rangle + \frac{L}{2}\mathbb{E}\|\mu_{t+1} - \mu_t\|^2 \tag{E.3}$$

$$= \mathbb{E}[f(\mu_t)] + \sum_{i=1}^{n}\sum_{j=1}^{n}\frac{1}{n^2}\mathbb{E}\langle\nabla f(\mu_t), -\frac{\eta}{n}\widetilde{g}_i(X_t^i) - \frac{\eta}{n}\widetilde{g}_j(X_t^j)\rangle \tag{E.4}$$

$$+ \sum_{i=1}^{n}\sum_{j=1}^{n}\frac{L}{2n^2}\mathbb{E}\|\frac{\eta}{n}\widetilde{g}_i(X_t^i) + \frac{\eta}{n}\widetilde{g}_j(X_t^j)\|^2 \tag{E.5}$$

$$\leq \mathbb{E}[f(\mu_t)] + \sum_{i=1}^{n}\sum_{j=1}^{n}\frac{1}{n^2}\mathbb{E}\langle\nabla f(\mu_t), -\frac{\eta}{n}\widetilde{g}_i(X_t^i) - \frac{\eta}{n}\widetilde{g}_j(X_t^j)\rangle \tag{E.6}$$

$$+ \sum_{i=1}^{n}\sum_{j=1}^{n}\frac{L}{n^2}\mathbb{E}\Big[\|\frac{\eta}{n}\widetilde{g}_i(X_t^i)\|^2 + \|\frac{\eta}{n}\widetilde{g}_j(X_t^j)\|^2\Big] \tag{E.7}$$

$$= \mathbb{E}[f(\mu_t)] + \sum_{i=1}^{n}\frac{2}{n}\mathbb{E}\langle\nabla f(\mu_t), -\frac{\eta}{n}\widetilde{g}_i(X_t^i)\rangle + \sum_{i=1}^{n}\frac{2L}{n}\mathbb{E}\|\frac{\eta}{n}\widetilde{g}_i(X_t^i)\|^2. \tag{E.8}$$

Using $\mathbb{E}[\widetilde{g}_i(x)] = \nabla f(x)$ and property (2.5) we can rewrite the above inequality as

$$\mathbb{E}[f(\mu_{t+1})] \leq \mathbb{E}[f(\mu_t)] + \sum_{i=1}^{n}\frac{2}{n}\mathbb{E}\langle\nabla f(\mu_t), -\frac{\eta}{n}\nabla f(X_t^i)\rangle + \sum_{i=1}^{n}\frac{2L\eta^2}{n^3}M^2 \tag{E.9}$$

$$= \mathbb{E}[f(\mu_t)] + \sum_{i=1}^{n}\frac{2\eta}{n^2}\mathbb{E}\langle\nabla f(\mu_t), \nabla f(\mu_t) - \nabla f(X_t^i)\rangle \tag{E.10}$$

$$- \sum_{i=1}^{n}\frac{2\eta}{n^2}\mathbb{E}\|\nabla f(\mu_t)\|^2 + \sum_{i=1}^{n}\frac{2L\eta^2}{n^3}M^2 \tag{E.11}$$

$$\leq \mathbb{E}[f(\mu_t)] + \sum_{i=1}^{n}\frac{\eta}{n^2}\mathbb{E}\Big[\|\nabla f(\mu_t)\|^2 + \|\nabla f(\mu_t) - \nabla f(X_t^i)\|^2\Big] \tag{E.12}$$

$$- \sum_{i=1}^{n}\frac{2\eta}{n^2}\mathbb{E}\|\nabla f(\mu_t)\|^2 + \sum_{i=1}^{n}\frac{2L\eta^2}{n^3}M^2 \tag{E.13}$$

$$= \mathbb{E}[f(\mu_t)] + \sum_{i=1}^{n}\frac{\eta}{n^2}\mathbb{E}\|\nabla f(\mu_t) - \nabla f(X_t^i)\|^2 - \sum_{i=1}^{n}\frac{\eta}{n^2}\mathbb{E}\|\nabla f(\mu_t)\|^2 + \sum_{i=1}^{n}\frac{2L\eta^2}{n^3}M^2 \tag{E.14}$$

$$\overset{L-smoothness}{\leq} \mathbb{E}[f(\mu_t)] + \sum_{i=1}^{n}\frac{L^2\eta}{n^2}\mathbb{E}\|\mu_t - X_t^i\|^2 - \frac{\eta}{n}\mathbb{E}\|\nabla f(\mu_t)\|^2 + \frac{2L\eta^2}{n^2}M^2. \tag{E.15}$$

recall that by Lemma E.1 we have that $\mathbb{E}[\Gamma_t] = \sum_{i=1}^{n}\mathbb{E}\|\mu_t - X_t^i\|^2 \leq 36n\eta^2 M^2$, hence the above inequality becomes:

$$\mathbb{E}[f(\mu_{t+1})] - \mathbb{E}[f(\mu_t)] \leq \frac{36L^2\eta^3 M^2}{n} - \frac{\eta}{n}\mathbb{E}\|\nabla f(\mu_t)\|^2 + \frac{2L\eta^2}{n^2}M^2. \tag{E.16}$$

by summing the above inequality for $t = 0$ to $t = T - 1$, we get that

$$\mathbb{E}[f(\mu_T)] - f(\mu_0) \leq \sum_{t=0}^{T-1}\Big(\frac{36L^2\eta^3 M^2}{n} - \frac{\eta}{n}\mathbb{E}\|\nabla f(\mu_t)\|^2 + \frac{2L\eta^2}{n^2}M^2\Big). \tag{E.17}$$

From this we get that :

$$\sum_{t=0}^{T-1} \frac{\eta}{n} \mathbb{E}\|\nabla f(\mu_t)\|^2 \le f(\mu_0) - \mathbb{E}[f(\mu_T)] + \sum_{t=0}^{T-1} \frac{36L^2\eta^3 M^2}{n} + \sum_{t=0}^{T-1} \frac{2L\eta^2}{n^2} M^2. \tag{E.18}$$

Note that $\mathbb{E}[f(\mu_T)] \ge f(x^*)$, hence after multiplying the above inequality by $\frac{n}{\eta T}$ we get that

$$\frac{1}{T} \sum_{t=0}^{T-1} \mathbb{E}\|\nabla f(\mu_t)\|^2 \le \frac{n(f(\mu_0) - f(x^*))}{T\eta} + 36LM^2\eta^2 + \frac{2LM^2\eta}{n}.$$

Observe that $\eta = n/\sqrt{T} \le 1/n$, since $T \ge n^4$. This allows us to finish the proof:

$$\frac{1}{T} \sum_{t=0}^{T-1} \mathbb{E}\|\nabla f(\mu_t)\|^2 \le \frac{n(f(\mu_0) - f(x^*))}{T\eta} + \frac{36LM^2\eta}{n} + \frac{2LM^2\eta}{n}$$

$$= \frac{(f(\mu_0) - f(x^*))}{\sqrt{T}} + \frac{36LM^2}{\sqrt{T}} + \frac{2LM^2}{\sqrt{T}}.$$

$\square$

Next we replace assumption 2.5 with assumption 2.4. We start by proving the following lemma:

**Lemma E.2.** *For any time step $t$ , we have:*

$$\mathbb{E}[\Delta_t|X_t] \le -\frac{\Gamma_t}{2n} + \frac{4\eta^2}{n} \sum_{i=1}^{n} \mathbb{E}\|\widetilde{g}_i(X_t^i)\|^2.$$

*Proof.* First we bound change in potential $\Delta_t = \Gamma_{t+1} - \Gamma_t$ for some time step $t > 0$. Let $\Delta_t^{i,j}$ be a change in potential when we choose different agents $i$ and $j$ at random and let $\Delta_t^i$ be a change in potential when we select the same node $i$. We get that

$$\mathbb{E}\Big[\Delta_t|X_t\Big] = \sum_i \sum_{i \ne j} \frac{1}{n^2} \mathbb{E}\Big[\Delta_t^{i,j}|X_t\Big] + \sum_{i=1}^{n} \frac{1}{n^2} \mathbb{E}\Big[\Delta_t^i|X_t\Big]. \tag{E.19}$$

We proceed by bounding a change in potential for fixed $i \ne j$. Observe, that in this case $\mu_{t+1} = \mu_t - (\eta\widetilde{g}_i(X_t^i) + \eta\widetilde{g}_j(X_t^j))/n$ and $X_{t+1}^i = X_{t+1}^j = (X_t^i + X_t^j)/2 - (\eta\widetilde{g}_i(X_t^i) + \eta\widetilde{g}_j(X_t^j))/2$. Hence,

$$X_{t+1}^i - \mu_{t+1} = X_{t+1}^j - \mu_{t+1} = (X_t^i + X_t^j)/2 - \frac{n-2}{2n}(\eta\widetilde{g}_i(X_t^i) + \eta\widetilde{g}_j(X_t^j)) - \mu_t.$$

For $k \notin \{i,j\}$, since $X_{t+1}^k = X_t^k$ we get that

$$X_{t+1}^k - \mu_{t+1} = X_t^k + \frac{1}{n}(\eta\widetilde{g}_i(X_t^i) + \eta\widetilde{g}_j(X_t^j)) - \mu_t.$$

This gives us that

$$\begin{aligned}
\mathbb{E}\Big[\Delta_t^{i,j}|X_t\Big] = &\ \mathbb{E}\Big\|(X_t^i + X_t^j)/2 - \frac{n-2}{2n}(\eta\widetilde{g}_i(X_t^i) + \eta\widetilde{g}_j(X_t^j)) - \mu_t\Big\|^2 - \|X_t^i - \mu_t\|^2 \\
&+ \mathbb{E}\Big\|(X_t^i + X_t^j)/2 - \frac{n-2}{2n}(\eta\widetilde{g}_i(X_t^i) + \eta\widetilde{g}_j(X_t^j)) - \mu_t\Big\|^2 - \|X_t^j - \mu_t\|^2 \\
&+ \sum_{k \notin \{i,j\}} \Big(\mathbb{E}\Big\|X_t^k + \frac{1}{n}(\eta\widetilde{g}_i(X_t^i) + \eta\widetilde{g}_j(X_t^j)) - \mu_t\Big\|^2 - \|X_t^k - \mu_t\|^2\Big) \\
= &\ 2\|(X_t^i - \mu_t)/2 + (X_t^j - \mu_t)/2\|^2 - \|X_t^i - \mu_t\|^2 - \|X_t^j - \mu_t\|^2 \\
&- \frac{n-2}{n}\mathbb{E}\langle\eta\widetilde{g}_i(X_t^i) + \eta\widetilde{g}_j(X_t^j), (X_t^i - \mu_t) + (X_t^j - \mu_t)\rangle \\
&+ 2\Big(\frac{n-2}{2n}\Big)^2 \mathbb{E}\|\eta\widetilde{g}_i(X_t^i) + \eta\widetilde{g}_j(X_t^j)\|^2 \\
&+ \sum_{k \notin \{i,j\}} \Big(\frac{2}{n}\mathbb{E}\langle\eta\widetilde{g}_i(X_t^i) + \eta\widetilde{g}_j(X_t^j), X_t^k - \mu_t\rangle + \frac{1}{n^2}\mathbb{E}\|\eta\widetilde{g}_i(X_t^i) + \eta\widetilde{g}_j(X_t^j)\|^2\Big)
\end{aligned}$$

Observe that

$$\sum_{k=1}^{n} \mathbb{E}\langle \eta \widetilde{g}_i(X_t^i) + \eta \widetilde{g}_j(X_t^j), X_t^k - \mu_t\rangle = 0.$$

Thus, we have that

$$
\begin{aligned}
\mathbb{E}\left[\Delta_t^{i,j}|X_t\right] \leq \quad & 2\|(X_t^i - \mu_t)/2 + (X_t^j - \mu_t)/2\|^2 - \|X_t^i - \mu_t\|^2 - \|X_t^j - \mu_t\|^2 \\
& - \mathbb{E}\langle \eta \widetilde{g}_i(X_t^i) + \eta \widetilde{g}_j(X_t^j), (X_t^i - \mu_t) + (X_t^j - \mu_t)\rangle \\
& + 4\eta^2 \left(\frac{n-2}{2n}\right)^2 (\mathbb{E}\|\widetilde{g}_i(X_t^i)\|^2 + \mathbb{E}\|\widetilde{g}_j(X_t^j)\|^2) + \sum_{k\notin\{i,j\}} \frac{2\eta^2}{n^2}(\mathbb{E}\|\widetilde{g}_i(X_t^i)\|^2 + \mathbb{E}\|\widetilde{g}_j(X_t^j)\|^2) \\
\leq \quad & -\|X_t^i - \mu_t\|^2/2 - \|X_t^j - \mu_t\|^2/2 + \langle X_t^i - \mu_t, X_t^j - \mu_t\rangle \\
& - \mathbb{E}\langle \eta \widetilde{g}_i(X_t^i) + \eta \widetilde{g}_j(X_t^j), (X_t^i - \mu_t) + (X_t^j - \mu_t)\rangle \\
& + \eta^2 (\mathbb{E}\|\widetilde{g}_i(X_t^i)\|^2 + \mathbb{E}\|\widetilde{g}_j(X_t^j)\|^2).
\end{aligned}
\tag{E.21}
$$

similarly we can prove that

$$\mathbb{E}\left[\Delta_t^i|X_t\right] \leq -\mathbb{E}\langle \eta \widetilde{g}_i(X_t^i) + \eta \widetilde{g}_i(X_t^i), (X_t^i - \mu_t) + (X_t^i - \mu_t)\rangle + 2\eta^2 (\mathbb{E}\|\widetilde{g}_i(X_t^i)\|^2). \tag{E.22}$$

By using inequalities E.21 and E.22 in inequality E.19 we get that

$$
\begin{aligned}
\mathbb{E}\left[\Delta_t|X_t\right] = \quad & \sum_i \sum_{i\neq j} \frac{1}{n^2} \mathbb{E}\left[\Delta_t^{i,j}|X_t\right] + \sum_{i=1}^{n} \frac{1}{n^2} \mathbb{E}\left[\Delta_i^t|X_t\right] \\
\leq \quad & -\sum_i \sum_{i\neq j} \frac{1}{n^2}\left(\|X_t^i - \mu_t\|^2/2 + \|X_t^j - \mu_t\|^2/2\right) + \sum_i \sum_{i\neq j} \frac{1}{n^2}\langle X_t^i - \mu_t, X_t^j - \mu_t\rangle \\
& -\sum_i \sum_j \frac{1}{n^2}\mathbb{E}\langle \eta \widetilde{g}_i(X_t^i) + \eta \widetilde{g}_j(X_t^j), (X_t^i - \mu_t) + (X_t^j - \mu_t)\rangle + \frac{2\eta^2}{n}\sum_{i=1}^{n} \mathbb{E}\|\widetilde{g}_i(X_t^i)\|^2.
\end{aligned}
$$

Observe that

$$\sum_i \sum_{i\neq j} \frac{1}{n^2}\langle X_t^i - \mu_t, X_t^j - \mu_t\rangle = \sum_{i=1}^{n} \frac{1}{n^2}\langle X_t^i - \mu_t, \sum_{j\neq i} X_t^j - \mu_t\rangle = \frac{1}{n^2}\sum_i -\|X_t^i - \mu_t\|^2 = -\frac{1}{n^2}\Gamma_t.$$

and

$$\sum_i \sum_{i\neq j} \frac{1}{n^2}\left(\|X_t^i - \mu_t\|^2/2 + \|X_t^j - \mu_t\|^2/2\right) = \frac{n-1}{n^2}\sum_i \|X_t^i - \mu_t\|^2 = \frac{n-1}{n^2}\Gamma_t.$$

Hence, we get that

$$\mathbb{E}\left[\Delta_t|X_t\right] \leq -\frac{\Gamma_t}{n} - \sum_i \sum_j \frac{1}{n^2}\mathbb{E}\langle \eta \widetilde{g}_i(X_t^i) + \eta \widetilde{g}_j(X_t^j), (X_t^i - \mu_t) + (X_t^j - \mu_t)\rangle + \frac{2\eta^2}{n}\sum_{i=1}^{n} \mathbb{E}\|\widetilde{g}_i(X_t^i)\|^2.$$
$$\tag{E.24}$$

Further, we have that

$$\sum_i \sum_j \frac{1}{n^2}\mathbb{E}\langle \eta \widetilde{g}_i(X_t^i) + \eta \widetilde{g}_j(X_t^j), (X_t^i - \mu_t) + (X_t^j - \mu_t)\rangle$$

$$= \sum_i \sum_j \frac{1}{n^2}\mathbb{E}\langle \eta \widetilde{g}_i(X_t^i), (X_t^j - \mu_t)\rangle + \sum_i \sum_j \frac{1}{n^2}\mathbb{E}\langle \eta \widetilde{g}_j(X_t^j), (X_t^i - \mu_t)\rangle + \sum_{i=1}^{n} \frac{2\eta}{n}\mathbb{E}\langle \widetilde{g}_i(X_t^i), X_t^i - \mu_t\rangle$$

$$= \sum_{i=1}^{n} \frac{2\eta}{n}\mathbb{E}\langle \widetilde{g}_i(X_t^i), X_t^i - \mu_t\rangle \leq \sum_{i=1}^{n}\left(\frac{2\eta^2}{n}\mathbb{E}\|\widetilde{g}_i(X_t^i)\|^2 + \frac{1}{2n}\|X_t^i - \mu_t\|^2\right) = \frac{2\eta_t^2}{n}\sum_{i=1}^{n} \mathbb{E}\|\widetilde{g}_i(X_t^i)\|^2 + \frac{\Gamma_t}{2n}.$$

By plugging above inequality in inequality E.24, we get that

$$\mathbb{E}[\Delta_t|X_t] \leq -\frac{\Gamma_t}{2n} + \frac{4\eta^2}{n}\sum_{i=1}^{n} \mathbb{E}\|\widetilde{g}_i(X_t^i)\|^2.$$

Hence, considering the definition of $\Delta_t$ we get the proof of the Lemma.

$\square$

**Lemma E.3.**

$$\sum_{i=1}^{n} \mathbb{E}\|\widetilde{g}_i(X_t^i)\|^2 \le 2n\sigma^2 + 12L\mathbb{E}[\Gamma_t] + 6n\mathbb{E}\|\sum_{i=1}^{n} \nabla f(X_t^i)/n\|^2 \tag{E.25}$$

*Proof.*

$$\sum_{i=1}^{n} \mathbb{E}\|\widetilde{g}_i(X_t^i)\|^2 = \sum_{i=1}^{n} \mathbb{E}\|\widetilde{g}_i(X_t^i) - \nabla f(X_t^i) + \nabla f(X_t^i)\|^2$$

$$\le 2\sum_{i=1}^{n} \mathbb{E}\|\widetilde{g}_i(X_t^i) - \nabla f(X_t^i)\|^2 + 2\sum_{i=1}^{n} \mathbb{E}\|\nabla f(X_t^i)\|^2$$

$$= 2\sum_{i=1}^{n} \mathbb{E}\|\widetilde{g}_i(X_t^i) - \nabla f(X_t^i)\|^2$$

$$+ 2\sum_{i=1}^{n} \mathbb{E}\|\nabla f(X_t^i) - \nabla f(\mu_t) + \nabla f(\mu_t) - \sum_{j=1}^{n} \nabla f(X_t^j)/n + \sum_{j=1}^{n} \nabla f(X_t^j)/n\|^2$$

$$\le 2\sum_{i=1}^{n} \mathbb{E}\|\widetilde{g}_i(X_t^i) - \nabla f(X_t^i)\|^2 + 6\sum_{i=1}^{n} \mathbb{E}\|\nabla f(X_t^i) - \nabla f(\mu_t)\|^2 + 6\sum_{i=1}^{n} \mathbb{E}\|\sum_{j=1}^{n}(\nabla f(\mu_t) - \nabla f(X_t^j))/n\|^2$$

$$+ 6\sum_{i=1}^{n} \mathbb{E}\|\sum_{j=1}^{n} \nabla f(X_t^j)/n\|^2$$

$$\le 2n\sigma^2 + 6L\sum_{i=1}^{n} \mathbb{E}\|\mu_t - X_t^i\|^2 + \frac{6L}{n}\sum_{i=1}^{n}\sum_{j=1}^{n} \mathbb{E}\|\mu_t - X_t^j\|^2 + 6n\mathbb{E}\|\sum_{i=1}^{n} \nabla f(X_t^i)/n\|^2$$

$$= 2n\sigma^2 + 12L\mathbb{E}[\Gamma_t] + 6n\mathbb{E}\|\sum_{i=1}^{n} \nabla f(X_t^i)/n\|^2.$$

$\square$

Notice that for $\eta \le \frac{1}{16\sqrt{L}}$ the above two lemmas give us

$$\mathbb{E}[\Gamma_{t+1}] \le (1 - \frac{1}{4n})\mathbb{E}[\Gamma_t] + 8\eta^2\sigma^2 + 24\eta^2\mathbb{E}\|\sum_{i=1}^{n} \nabla f(X_t^i)/n\|^2. \tag{E.26}$$

Observe that since $\sum_{i=0}^{\infty}(1 - 1/4n)^i = 4n$, the above equation results in

**Lemma E.4.**

$$\sum_{t=0}^{T} \mathbb{E}[G_t] \le 32Tn\eta^2\sigma^2 + \sum_{t=0}^{T-1} 96n\eta^2\mathbb{E}\|\sum_{i=1}^{n} \nabla f(X_t^i)/n\|^2 \tag{E.27}$$

Now, we are ready to prove the following theorem

**Theorem 5.3.** *Let $f$ be an non-convex, L-smooth, function satisfying condition 2.4, whose minimum $x^\star$ we are trying to find via the PopSGD procedure given in Algorithm 1. Let the learning rate we use be $\eta = n/\sqrt{T}$. Then, for any $T \ge 4624 \max\{1/L^2, 1\}n^4$, we have*

$$\sum_{t=0}^{T-1} \frac{1}{T}\mathbb{E}\|f(\mu_t)\|^2 \le \frac{\mathbb{E}[f(\mu_0)] - \mathbb{E}[f(x^\star)]}{\sqrt{T}} + \frac{4\sigma^2}{\sqrt{T}} + \frac{32L^2\sigma^2}{\sqrt{T}} + \frac{768L^2\sigma^2}{\sqrt{T}}.$$

*Proof.*

$$\mathbb{E}[f(\mu_{t+1})] \overset{L-smoothness}{\le} \mathbb{E}[f(\mu_t)] + \mathbb{E}\langle\nabla f(\mu_t), \mu_{t+1} - \mu_t\rangle + \frac{L}{2}\mathbb{E}\|\mu_{t+1} - \mu_t\|^2$$

$$\begin{aligned}
&= \mathbb{E}[f(\mu_t)] + \sum_{i=1}^{n}\sum_{j=1}^{n}\frac{1}{n^2}\mathbb{E}\langle\nabla f(\mu_t), -\frac{\eta}{n}\widetilde{g}_i(X_t^i) - \frac{\eta}{n}\widetilde{g}_j(X_t^j)\rangle \\
&\quad + \sum_{i=1}^{n}\sum_{j=1}^{n}\frac{L}{2n^2}\mathbb{E}\|\frac{\eta}{n}\widetilde{g}_i(X_t^i) + \frac{\eta}{n}\widetilde{g}_j(X_t^j)\|^2 \\
&\leq \mathbb{E}[f(\mu_t)] + \sum_{i=1}^{n}\sum_{j=1}^{n}\frac{1}{n^2}\mathbb{E}\langle\nabla f(\mu_t), -\frac{\eta}{n}\widetilde{g}_i(X_t^i) - \frac{\eta}{n}\widetilde{g}_j(X_t^j)\rangle \\
&\quad + \sum_{i=1}^{n}\sum_{j=1}^{n}\frac{L}{n^2}\mathbb{E}\Big[\|\frac{\eta}{n}\widetilde{g}_i(X_t^i)\|^2 + \|\frac{\eta}{n}\widetilde{g}_j(X_t^j)\|^2\Big] \\
&= \mathbb{E}[f(\mu_t)] + \sum_{i=1}^{n}\frac{2}{n}\mathbb{E}\langle\nabla f(\mu_t), -\frac{\eta}{n}\widetilde{g}_i(X_t^i)\rangle + \sum_{i=1}^{n}\frac{2L}{n}\mathbb{E}\|\frac{\eta}{n}\widetilde{g}_i(X_t^i)\|^2 \\
&= \mathbb{E}[f(\mu_t)] + \sum_{i=1}^{n}\frac{2}{n}\mathbb{E}\langle\nabla f(\mu_t), -\frac{\eta}{n}\nabla f(X_t^i)\rangle + \sum_{i=1}^{n}\frac{2L}{n}\mathbb{E}\|\frac{\eta}{n}\widetilde{g}_i(X_t^i)\|^2 \\
&= \mathbb{E}[f(\mu_t)] + \frac{\eta}{n}\mathbb{E}\|\sum_{i=1}^{n}(\nabla f(\mu_t) - \nabla f(X_t^i))/n\|^2 - \frac{\eta}{n}\mathbb{E}\|\nabla f(\mu_t)\|^2 \\
&\quad - \frac{\eta}{n}\mathbb{E}\|\sum_{i=1}^{n}\nabla f(X_t^i)/n\|^2 + \sum_{i=1}^{n}\frac{2L}{n}\mathbb{E}\|\frac{\eta}{n}\widetilde{g}_i(X_t^i)\|^2 \\
&\leq \mathbb{E}[f(\mu_t)] + \frac{\eta L^2}{n^2}\mathbb{E}[\Gamma_t] - \frac{\eta}{n}\mathbb{E}\|\nabla f(\mu_t)\|^2 - \frac{\eta}{n}\mathbb{E}\|\sum_{i=1}^{n}\nabla f(X_t^i)/n\|^2 \\
&\quad + \frac{4\sigma^2\eta^2}{n^2} + \frac{24L^2\eta^2}{n^3}\mathbb{E}[\Gamma_t] + \frac{12L\eta^2}{n^2}\mathbb{E}\|\sum_{i=1}^{n}\nabla f(X_t^i)/n\|^2.
\end{aligned}$$

Next, we sum the above inequality from $t = 0$ to $t = T - 1$ and divide by $T$. We get:

$$\begin{aligned}
\sum_{t=0}^{T-1}\frac{\eta}{Tn}\mathbb{E}\|f(\mu_t)\|^2 &\leq \frac{\mathbb{E}[f(\mu_0)] - \mathbb{E}[f(\mu_T)]}{T} + \frac{4\sigma^2\eta^2}{n^2} + \sum_{t=0}^{T-1}\frac{\eta L^2}{n^2 T}\mathbb{E}[\Gamma_t] + \sum_{t=0}^{T-1}\frac{24\eta^2 L^2}{n^3 T}\mathbb{E}[\Gamma_t] \\
&\quad + \sum_{t=0}^{T-1}\frac{12L\eta^2}{n^2 T}\mathbb{E}\|\sum_{i=1}^{n}\nabla f(X_t^i)/n\|^2 - \sum_{t=0}^{T-1}\frac{\eta}{nT}\mathbb{E}\|\sum_{i=1}^{n}\nabla f(X_t^i)/n\|^2 \\
&\overset{\text{Lemma E.4}}{\leq} \frac{\mathbb{E}[f(\mu_0)] - \mathbb{E}[f(\mu_T)]}{T} + \frac{4\sigma^2\eta^2}{n^2} + \frac{32\eta^3 L^2\sigma^2}{n} + \sum_{t=0}^{T-1}\frac{96\eta^3 L^2}{nT}\mathbb{E}\|\sum_{i=1}^{n}\nabla f(X_t^i)/n\|^2 \\
&\quad + \frac{768\eta^4 L^2\sigma^2}{n^2} + \sum_{t=0}^{T-1}\frac{2304\eta^4 L^2}{n^2 T}\mathbb{E}\|\sum_{i=1}^{n}\nabla f(X_t^i)/n\|^2 \\
&\quad - \sum_{t=0}^{T-1}\frac{\eta}{nT}\mathbb{E}\|\sum_{i=1}^{n}\nabla f(X_t^i)/n\|^2.
\end{aligned}$$

Next we assume that $\eta < \frac{1}{68L}$ and $\eta < 1$. This allows us to eliminate terms with $\mathbb{E}\|\sum_{i=1}^{n}\nabla f(X_t^i)/n\|^2$ multiplicative factor from the above inequality, hence we get:

$$\sum_{t=0}^{T-1}\frac{\eta}{Tn}\mathbb{E}\|f(\mu_t)\|^2 \leq \frac{\mathbb{E}[f(\mu_0)] - \mathbb{E}[f(\mu_T)]}{T} + \frac{4\sigma^2\eta^2}{n^2} + \frac{32\eta^3 L^2\sigma^2}{n} + \frac{768\eta^4 L^2\sigma^2}{n^2}.$$

next, we divide the above inequality by $\frac{\eta}{n}$:

$$\sum_{t=0}^{T-1}\frac{1}{T}\mathbb{E}\|f(\mu_t)\|^2 \leq \frac{n(\mathbb{E}[f(\mu_0)] - \mathbb{E}[f(\mu_T)])}{T\eta} + \frac{4\sigma^2\eta}{n} + 32\eta^2 L^2\sigma^2 + \frac{768\eta^3 L^2\sigma^2}{n}.$$

Further, assuming that $\eta \leq 1/n$, we get:

$$
\begin{aligned}
\sum_{t=0}^{T-1} \frac{1}{T} \mathbb{E}\|f(\mu_t)\|^2 &\leq \frac{n(\mathbb{E}[f(\mu_0)] - \mathbb{E}[f(\mu_T)])}{T\eta} + \frac{4\sigma^2\eta}{n} + \frac{32\eta L^2\sigma^2}{n} + \frac{768\eta L^2\sigma^2}{n} \\
&\leq \frac{n(\mathbb{E}[f(\mu_0)] - \mathbb{E}[f(x^*)])}{T\eta} + \frac{4\sigma^2\eta}{n} + \frac{32\eta L^2\sigma^2}{n} + \frac{768\eta L^2\sigma^2}{n} \\
&= \frac{\mathbb{E}[f(\mu_0)] - \mathbb{E}[f(x^*)]}{\sqrt{T}} + \frac{4\sigma^2}{\sqrt{T}} + \frac{32L^2\sigma^2}{\sqrt{T}} + \frac{768L^2\sigma^2}{\sqrt{T}}.
\end{aligned}
$$

Observe that all the assumptions on $\eta$ are satisfied for $T \geq 4624 \max\{1/L^2, 1\} n^4$. $\qquad\square$

## F   GENERAL INTERACTION GRAPHS

The crucial difference between interactions on arbitrary graph and interactions on clique graph comes from the fact that we get different bound on Gamma potential.

In the case of arbitrary graph we can show that:

$$
\mathbb{E}[\Gamma_{t+1}|\Gamma_t] \leq \left(1 - \Theta(\lambda_2/m)\right)\Gamma_t + 4\eta_t M\left(\frac{\Gamma_t}{n}\right)^{1/2} + 8\eta_t^2 M^2.
$$

Where $\lambda_2$ is a second smallest eigenvalue of the Laplacian of the interaction graph and $m$ is the number of it's edges. This allows us to bound $\Gamma$ and then we can follow analysis of clique case to get the convergence rate. For example, in a case of cycle

$$
\mathbb{E}[\Gamma_{t+1}|\Gamma_t] \leq \left(1 - \Theta(1/n^3)\right)\Gamma_t + 4\eta_t M\left(\frac{\Gamma_t}{n}\right)^{1/2} + 8\eta_t^2 M^2.
$$

and hence

$$
\mathbb{E}[\Gamma_t] \leq O(n^5 \eta_t^2 M^2).
$$

and this allows us to prove Theorem 5.1.

