# OpenReview forum: "PopSGD: Decentralized Stochastic Gradient Descent in the Population Model"
_ICLR.cc/2020/Conference — Reject_

### Official Review · AnonReviewer3 · 2019-10-23
**Official Blind Review #3**

**Rating:** 3

**Review:**

The paper considers scaling distributed stochastic gradient descent to large number of nodes.  Paper proposes novel asynchronous variant to decentralized SGD, called PopSGD. It models asynchrony with the population model. Paper theoretically analyzes the proposed method and shows that in the convex case PopSGD has a linear speedup in the number of nodes compared to the sequential training on one node. However, in the non-convex setting, the PopSGD rate doesn’t have a linear speedup. The paper also provides experimental evaluation of PopSGD where they scale PopSGD up to 1000 of nodes in the convex optimization; and also apply PopSGD to neural network training on ImageNet.

My score is weak reject. The major reason is that it is not clear how does this work theoretically and experimentally compares to the previous asynchronous variants of decentralized SGD (Lian et al. (2017b) AD-PSGD and Assran et al. (2018) SGP) or to the centralized SGD (parallel mini-batch SGD) baseline; and what are the benefits of the proposed method.

Concerns that should be addressed:

1. No theoretical and experimental comparison with the baselines (see above).

2. Extension to the non-convex case: there is no linear speedup in the number of nodes n. Does this mean that it is better to use Centralized SGD (which has speedup in the number of nodes)? The comparison should be made explicit.


3. The paper is a bit too long (10 pages) and contains some repetitions. Consider to shorten a bit. (e.g. procedure of splitting data between nodes was described twice on page 2 and 4; proof overview on page 6 could be merged with two steps on page 7)


4. The procedure how to sample nodes uniformly was not discussed in the paper. Moreover, it is also not clear why \Theta(n) updates could be done in parallel. When intersection happens, many nodes would have to wait for the previously selected pairs to finish computation.


5. Why are the local learning rates required? When scheduler samples the nodes uniformly it can also transmit them the global time count.


6. In the description of data distribution (page 2, paragraph 4-5, page 4, last paragraph): what if there is more samples than available nodes? do the nodes exchange samples or only gradients? Is it possible to have part of the full dataset on every node without sharing it with anyone?


7. Page 3, line 3: Lian et al. (2017b) and Assran et al. (SGP) also showed that they don’t require global synchronization.


8. Page 4, line 2: The AD-PSGD rate does have a linear speedup in the number of workers n, so the claim should be corrected.


9. Page 7: all lemmas and theorems hold only for the global stepsize \eta_t. Would it also hold with local stepsizes \eta_t^i?


10. Extensions for arbitrary graphs: would it be possible to have one theorem for all possible graphs and see how graph parameters (e.g. spectral gap or others) influence the convergence rate?


11. Experiments: I didn’t understand the definition of mult constant and what does it control. Re-phrase this paragraph.

Minor comments:
- page 1, last line of the paragraph 2: “parameter obtained by node i at time t“ -> “stoch. gradient obtained by node i at time t“?
- page 2, line 4: “variants(e.g.” -> “variants (e.g.”
- Usually \mu is used for strong convexity parameter.
- page 3, paragraph 2, “we emphasize that convexity is not enough…” the purpose of this sentence is unclear, what is enough then or why is that important to know?
- page 3, related work: Nedic at al. Nedic et al. (2017) -> Nedic et al. (2017). The same for the other citation.
- page 3, related work: PP model is not defined.
- How can PP model result in a multigraph? If two samples pairs have the same nodes, then they need to be processed sequentially, so it can be modeled with two graphs for different time steps.
- Population protocol model (page 4): “states store real numbers” -> can they store vectors instead?
page 5, estimating time and the learning rate section: what happens if the V^i is equal to V_j? Who updates its value?
- Figure 1(a) was not discussed at all in the text.
- Page 6, Notation and preliminaries: why it is required that T = O(poly n) is not explained.


**Experience Assessment:**

I have published one or two papers in this area.

**Review Assessment: Checking Correctness Of Derivations And Theory:**

I assessed the sensibility of the derivations and theory.

**Review Assessment: Checking Correctness Of Experiments:**

I assessed the sensibility of the experiments.

**Review Assessment: Thoroughness In Paper Reading:**

I read the paper at least twice and used my best judgement in assessing the paper.

---

> ### Author Response · Authors · 2019-11-10
> **Response to individual questions**
>
> Please see the main rebuttal text for answers to the main points raised. Below, we address the remaining questions.
>
> [R2]  Extension to the non-convex case: there is no linear speedup in the number of nodes n.
>
> We can obtain 1 / sqrt n speedup in the general non-convex case, and *linear* speedup under the PL condition. These extensions will be added to the next version.
>
> [R2] The paper is too long.
>
> We will do our best to follow the reviewer’s comments in shortening the paper.
>
> [R2] Sampling procedure and parallelism.
>
> In practice, the averaging step does not need to be necessarily performed synchronously. We will clarify this in the paper.
>
> [R2] Why are local learning rates required?
>
> Because nodes lack a common notion of time. There is no central scheduler which chooses them to interact: whenever some node completes its minibatch computation, it chooses another node uniformly at random and initiates an averaging step. This other node’s time step may not be the same as that of the initiator.
>
> [R2] In the description of data distribution (page 2, paragraph 4-5, page 4, last paragraph): what if there is more samples than available nodes? do the nodes exchange samples or only gradients? Is it possible to have part of the full dataset on every node without sharing it with anyone?
>
> The nodes can exchange either samples or gradients (the argument would work in either case). The dataset can also be partitioned, as long as the partitions satisfy the technical assumptions. We will clarify this point.
>
> [R2] AD-SGD and SGP do not require global synchronization.
>
> Please see the discussion in the main rebuttal.
>
> [R2] The AD-PSGD rate does have a linear speedup in the number of workers n, so the claim should be corrected.
>
> Please see the discussion in the main rebuttal.
>
> [R2] All lemmas and theorems hold only for the global stepsize \eta_t. Would it also hold with local stepsizes \eta_t^i?
>
> Yes, the results also hold with respect to local stepsizes--we omitted the additional indexing for brevity. We will clarify this point.
>
> [R2] Extensions for arbitrary graphs: would it be possible to have one theorem for all possible graphs and see how graph parameters (e.g. spectral gap or others) influence the convergence rate?
> Yes, we can provide general bounds in terms of the spectral gap. We will add the statement to the next version.
>
> [R2] I didn’t understand the definition of mult constant and what does it control. Re-phrase this paragraph.
>
> We apologize if this was not clear. We will add an example in the next version along the lines of the one provided in the general rebuttal above.
>
> We also thank the reviewer for their detailed comments, which we will implement in the next version.

---

### Official Review · AnonReviewer2 · 2019-10-23
**Official Blind Review #2**

**Rating:** 1

**Review:**

This paper proposes to use population algorithms as a mechanism for implementing distributed training of deep neural networks. The paper makes some claims about the relationship to previous work on (asynchronous) gossip algorithms that appear to be incorrect. In fact, the proposed PopSGD algorithm is very closely related to other methods in the literature, including AD-PSGD (Lian et al. 2017b) and SGP (Assran et al. 2018). I recommend it be rejected due to lack of novelty and missing connections to much related work.

The introduction (page 3) mentions that the "matrix characterization is not possible in the population model." Here the "matrix characterization" refers to the typical approach in which gossip algorithms (synchronous or asynchronous) are formulated and analyzed. I'd appreciate if the authors could elaborate on this claim. In the study of gossip algorithms, the organization of time into "global rounds" is purely for the sake of analysis; a global, synchronized clock is not required to implement these methods. In fact, the description of the setup appears to be very similar to the asynchronous time model described used to analyze "randomized gossip algorithms" (see the well-cited paper by Boyd, Ghosh, Prabhakar, and Shah). In the PopSGD case, the choice is simply to allow the complete graph (i.e., any agent can interact with any other agent) rather than restricting interactions of a given agent to be among a subset of the other agents (i.e., its neighbors).

Let me elaborate on the ways in which PopSGD is similar to AD-PSGD and SGP. PopSGD involves interactions between randomly drawn pairs of agents. The AD-PSGD algorithm of Lian et al. (2017b) also performs updates between pairs of agents drawn randomly at every step. The definition of the PopSGD interaction in (1.1) (or equivalently Alg 1) implies that when agents i and j interact, neither i nor j can interact with another agent until the current interaction completes. The main difference appears to be that in Lian et al. (2017b) agents are organized into a bipartite graph where $n/2$ nodes are "active" and initiate interactions with one of the other $n/2$ "passive" nodes (drawn randomly). This is done for practical reasons - to avoid deadlocks.

I also believe that PopSGD can be viewed as a particular instance of the overlap-SGP algorithm proposed in Assran et al. (2018). Overlap-SGP, the way it is described, makes use of one-directional interactions (agent i may receive and incorporate information from agent j without the reverse happening simultaneously). This was also introduced for practical reasons. It is possible for multiple interactions to happen simultaneously, and the pattern of iteractions may vary over time. There is nothing in the analysis, however, that prevents one from restricting to symmetric interactions, in which case one could recover the symmetric updates of PopSGD. To compensate for one-directional interactions, Overlap-SGP tracks an additional variable (the weight, or denominator). However, in the case where interactions are always symmetric as in PopSGD, the corresponding update matrices will always be doubly-stochastic, and in this case the weights are always equal to 1. Thus PopSGD really is identical to Overlap-SGP in this special restricted case where interactions are always pair-wise and symmetric. Moreover, Assran et al. (2018) prove that Overlap-SGP achieves a linear speedup in the smooth non-convex setting.

The experiments don't provide any comparison with other related methods, and the discussion in the introduction isn't sufficient to convince me that there are significant differences between these methods. In the experiments, I also wanted to ask about the mult constant. If it is really possible to achieve linear scaling, wouldn't one hope to be able to get away with mult=1?

The decreasing learning rate schedule used in the description and analysis of PopSGD seems very restrictive. Specifically, in the training of deep neural networks it is common to use much different learning rate schedules. Is it fundamentally not possible to do so with PopSGD-type models, or is it just a limitation of the current analysis approach (specifically for convex functions)? What learning rate scheme was used in the experiments?

Finally, the introduction (p3) emphasizes that it is the population gradient perspective, and the connection to load-balancing processes, which enable one to achieve linear scaling. I disagree with this statement. While I do agree that convexity alone is not sufficient, the key assumption made here (as well as in other work, such as that of Lian et al.), is that all agents draw gradients from the same distribution; i.e., that all agents have access to independent and identically distributed stochastic gradient oracles. In fact, this is stronger than the assumptions made in Lian et al. (2017a and 2017b), and Assran et al. (2018), where it is only assumed that the gradient oracles at each agent are similar, but not necessarily identical.

**Experience Assessment:**

I have published in this field for several years.

**Review Assessment: Checking Correctness Of Derivations And Theory:**

I assessed the sensibility of the derivations and theory.

**Review Assessment: Checking Correctness Of Experiments:**

I carefully checked the experiments.

**Review Assessment: Thoroughness In Paper Reading:**

I read the paper at least twice and used my best judgement in assessing the paper.

---

> ### Author Response · Authors · 2019-11-10
> **Response to individual questions**
>
> Please see the main rebuttal text for answers to the main points raised. Below, we address the remaining questions.
>
> [R1] The technique (the connection to load-balacing processes) is not what enables linear scaling, but the (strong) assumption that the gradient oracles are identical.
>
> This is not exactly the case. In fact, our analysis in the non-convex case would continue to work under the weaker assumptions of (Lian et al.), although the bounds would be slightly different. We will clarify the use of this assumption in the next version.
>
> [R1] Why are decreasing LR schedules necessary?
>
> It is indeed the case that the decreasing LR schedule we describe is necessary (and common) for the convex analysis, but can be generalized in the non-convex case.

---

> > ### Comment · AnonReviewer2 · 2019-11-15
> > **Not convinced about relationship to previous work**
> >
> > Thank you for your responses.
> >
> > I still do not agree with your assessment of the relationship to previous work. In (Lian et al.), reading the discussion of implementation details in Sec 3.3 and the description of wait-free continuous training and communication in the appendix, it is clear that the algorithm does not require to be implemented in lock-step rounds. Similarly, Assran et al. describe synchronous and asynchronous versions of their method. Specifically, an asynchronous "overlap" version of stochastic gradient push is described in Sec 3 of that paper, and the analysis in Sec 4 covers this asynchronous version of the method, facilitated by allowing a time-varying matrix P(k). I'm also still curious to hear if/how the population protocol model adopted in this paper relates to the asynchronous time model described in Boyd, Ghosh, Prabhakar, and Shah, "Randomized gossip algorithms".
> >
> > Regarding what assumption(s) enable linear scaling, the weaker assumption of Lian et al. still ensures that the objectives at different nodes are similar. If no such assumption is made (so that the objectives/data distributions at different nodes can be arbitrarily different), would PopSGD still obtain a linear speedup?

---

> > > ### Author Response · Authors · 2019-11-15
> > > **Reviewer Response**
> > >
> > > We sincerely thank the reviewer for engaging with us, and for the insightful questions.
> > > We address them below.
> > >
> > > We begin by addressing the question regarding the relationship to the paper of (Boyd et al.).
> > > This paper considers averaging via gossip in two models:
> > > 1) synchronous gossip, structured in global rounds, where each node interacts with a randomly chosen neighbor, and
> > > 2) asynchronous gossip, where each node wakes up at times given by a local Poisson clock, and picks a random neighbor to interact with.
> > > The population model is functionally equivalent to the asynchronous gossip model, since the interaction times in the latter model can be "discretized" to lead to pairwise uniform interactions. The key difference between our work and averaging in the gossip model is that their input model is static (node inputs are fixed, and node estimates must converge to the true mean), whereas we study the a dynamic setting, where the models are updated in each round by SGD, and should remain concentrated around the parameter mean as it converges towards the optimum.
> > > We have added this discussion to the revision.
> > >
> > > With this in mind, we address the following question:
> > >
> > > > In (Lian et al.), reading the discussion of implementation details in Sec 3.3 and the description of wait-free continuous training and communication in the appendix, it is clear that the algorithm does not require to be implemented in lock-step rounds. Similarly, Assran et al. describe synchronous and asynchronous versions of their method. Specifically, an asynchronous "overlap" version of stochastic gradient push is described in Sec 3 of that paper, and the analysis in Sec 4 covers this asynchronous version of the method, facilitated by allowing a time-varying matrix P(k).
> > >
> > > Both papers describe their baseline algorithms in terms of the synchronous gossip model (see previous answer): interactions are round-based, following a random or deterministic matching in every "round."
> > > For (Lian et al.), this is described in sections 3.2 and 3.3.1; for (Assran et al.) the details can be found in Section 3, and in Appendix A.
> > > Next, to avoid deadlocks/slowdown, both papers proceed to *relax* these synchrony requirements by allowing nodes to perform updates based on stale information: this is described in Appendix A of (Lian et al.), and in the Overlap-SGP section for (Assran et al.).
> > >
> > > Our key point here is that the resulting relaxed models are not identical to---and do not subsume---the population model/asynchronous gossip model.
> > > In the population model, each step is a uniform random pairwise interaction: in particular, due to randomness, it is possible for a node A to interact several times before some other node B interacts at all. In the synchronous gossip model, even with the consistency relaxation, nodes would have to interact once every "round," even if they may see stale information.
> > >
> > > Finally, a reasonable question is whether their analyses can be adapted to analyze the population model.
> > > As detailed in the previous response, this is possible for (Lian et al.), but would yield weaker bounds in the non-convex case compared to our analysis. (For instance, their bounds would only apply after $T \geq \Theta( n^6 )$ steps.)
> > > One could also try to analyze the PP model using the techniques of (Assran et al.), by building the interaction graph via the sequence of random interactions in the PP model. We believe this is similar to what the reviewer is suggesting.
> > > Unfortunately, in this case, the resulting bound would have no speedup relative to the sequential case (or even negative speedup): this is because the numerator in their convergence bound (the parameter C in Thm1) depends linearly on the diameter of the interaction graph (which must be connected). It is easy to see (e.g. by standard Erdoes-Renyi) that in the PP model this diameter would be linear in $n$. We note that this parameter C is also linear in $\sqrt d$, the model dimension.
> > >
> > > We therefore maintain the claim that our analysis yields the best bounds for the PP model, even in the non-convex case.
> > >
> > > > Regarding what assumption(s) enable linear scaling, the weaker assumption of Lian et al. still ensures that the objectives at different nodes are similar. If no such assumption is made ([...]), would PopSGD still obtain a linear speedup?
> > >
> > > Yes, SGD would still obtain linear speedup, but the assumptions and objective would have to be adjusted. More precisely, in this case, as in (Lian et al.), we would have to assume that the objective $f_i$ for each process $i$, is $L$-smooth and $\ell$-strongly convex. Additionally, we would need to assume bounded gradients for each objective.
> > >
> > > Let $f(x)=\sum_{i=1}^n f_i(x) / n$ be our global objective function. In this setting,  we will be able to achieve the same speedup, but not for the objective $f(\mu_T)-f(x^*)$.  Instead, we would get the covergence bound for $\sum_{i=1}^n ( f_i(X_T^i)-f(x*))/n$, where $X_T^i$ is the value of model $i$ at step $T$.

---

### Author Response · Authors · 2019-11-10
**Rebuttal regarding relation to previous work**

Dear reviewers,

Thank you for your reviews. We summarize your feedback, discuss it, and outline our planned changes below. We are working on implementing these changes now, and will provide an updated version within the next few days.
This rebuttal has two parts: the first addresses the relation previous work, whereas the second addresses questions regarding linear scaling and experiments. We also respond to individual comments separately.

It would be extremely useful to us if the reviewers would signal whether they agree with our comments and update plan for the draft.

Issue 1: our model and results are subsumed by previous work by (Lian et al.) and (Assran et al.).

This is not the case. As stated in the submission, the analytical models presented in the above papers are round-based: every node is assumed to interact exactly once in each communication round, in particular forming a perfect matching in every round.

This is clear in the model and algorithm descriptions in the papers. Please see e.g. line 5 of Algorithm 1 in (Lian et al.), version https://arxiv.org/pdf/1710.06952.pdf, which describes global averaging in each step; further, (Assran et al.) provide a detailed explanation of why they chose a deterministic perfect matching model, and not random edge sampling. We quote from (Assran et al.):
"[...] One could consider designing the matrices P(k) in a stochastic manner, where each node randomly samples one neighbor to send to at every iteration. [...But] random schemes are still not as effective at quickly averaging as deterministically cycling through neighbors in the directed exponential graph. Moreover, with randomized schemes, we are no longer guaranteed that each node receives the same number of messages at every iteration, so the communication load will not be balanced as in the deterministic scheme.”

We hope this establishes the fact that these papers consider global round-based matching models, and not uniform edge-sampling methods. This fact was stated in our original submission. This distinction is important, since it simplifies the algorithm, connects to a fundamental model in distributed computing, and allows for faster implementation.

The remaining question is whether the techniques of (Lian et al.) and (Assran et al.) could be *modified* to analyze PopSGD. We spent a considerable amount of time looking into this, given the reviewer comments.
Our conclusion is the techniques of (Lian et al.) could be adapted to analyze PopSGD in the *non-convex* case, but that the resulting bounds would be weaker, by a polynomial factor in n, the number of nodes. This polynomial difference is linked to the fact that a matrix characterization is used in their analysis.
More precisely, one can instantiate the matrix Wk to be the interaction matrix between only a random pair of nodes, and can relax Assumption 1.2 in the paper to state the Wk only needs to be doubly stochastic *in expectation*. On can then carefully follow through the rest of their argument; the resulting bound would be off by at least a n^2 factor from the bound we obtain on Gamma.
We were not able to apply the analysis technique of (Assran et al.) to obtain better bounds in our setting.

We note however that the above discussion has little bearing on the convex case, which is the main result of our submission. The convex case is not considered in these previous papers; in this case, we are the first to provide linear convergence speedup (see speedup discussion below).
We will provide a detailed explanation of these connections in the updated version of our draft.

---

### Author Response · Authors · 2019-11-10
**Rebuttal regarding linear speedup, hyperparameters, and experiments**

(This text is the second part of the main rebuttal text, which can be found below.)

Issue 2: linear speedup, the learning rate regime and hyperparameters.

We first clarify that by linear convergence speedup we mean that the time to convergence is divided by n. (This is what we guarantee in the convex case--please see Thm. 4.1 and its discussion.)
The bounds provided by (Lian et al.) and (Assran et al.) divide convergence time by sqrt{n}, assuming the rest of the parameters are constant. (We believe this is the best possible in the non-convex case barring additional assumptions.)

We are in fact able to obtain the same sqrt dependency on n in the non-convex case, as in the work of (Lian et al.). This is a minor modification to the current argument; we will update the draft to reflect this.

In fact, under the Polyak-Łojasiewicz condition, we can obtain *linear* speedup in the non-convex case as well. This new extension result will be added to the current submission.

The hyperparameter recipes we use for training (learning rate regime, momentum, batch size, etc.) are exactly the same as that of the standard sequential, but scaled by mult / num_nodes. This was stated in Section 6.
For example, if sequential SGD trains ResNet18 in 90 epochs, decreasing the learning rate at 30 and 60 epochs, then PopSGD with 20 nodes and multiplier 2 would use 90 * 2 / 20 =  9 epochs per node, decreasing the learning rate at 3 and 6 epochs. Thus, the total number of gradient evaluations is 2x larger than sequential, but the end-to-end time could be 10x less. We note that the addition of the multiplier parameter is justified by the theory, since the averaging procedure induces a “delay” which has to be compensated by additional iterations.
We note that this multiplier procedure is similar to that used by (Assran et al.), where they use the speedup of their algorithm relative to data-parallel as the multiplier value (see their section 6.2).

Issue 3: Comparisons with other algorithms, e.g. data-parallel SGD, D-SGD, AD-SGD, and SGP [R1, R2].

We did provide comparisons with data-parallel SGD and Local SGD: please see Section 6 and Appendix Section A. In particular, we showed that PopSGD converges faster in the convex case, and that it provides significant end-to-end speedup in the non-convex case as well versus both local SGD and data-parallel SGD (~2x end-to-end convergence speedup for ResNet 50). Please see Figure 4 in the original submission for the comparisons.
Regarding practical scaling, we recall that Figure 2 (3rd panel) exhibits almost linear scaling for PopSGD.

We are working on adding experiments for D-SGD and SGP.

---

### Author Response · Authors · 2019-11-15
**Revision Summary**

We again thank the reviewers for their feedback. We summarize our revision, and our claimed contributions.

- We have significantly re-written the introduction and related work, specifically for clarity with respect to the work of (Lian et al.) and (Assran et al.) We have added Table 1 (Appendix), which summarizes the assumptions and guarantees for both our work and previous algorithms.

- We maintain our claim that we are the first to prove linear speedup for convex objectives in the population protocol model.

- In addition, we proved linear speedup for PopSGD in the non-convex case under the PL assumption (a first for any decentralized algorithm), and $\sqrt n$ speedup in the classic non-convex case. In this latter case, our bounds are (significantly) stronger than those implied by previous work when applied to the population protocol model, under similar assumptions.

- Our results are based on a new fine-grained analysis technique, which enables us to provide better concentration for the individual models relative to the mean.

- We have added an extension to arbitrary graph topologies via the spectral gap (please see Appendix).

- We have added experimental results showing similar performance with respect to DA-SGD and SGP, and almost linear practical scaling. Comparisons to data-parallel SGD and Local SGD were also provided.

- We have performed a major overhaul of the paper to address all the reviewer comments.

---

### Decision · Program_Chairs · 2019-12-19

**Decision:**

Reject

**Comment:**

This manuscript studies scaling distributed stochastic gradient descent to a large number of nodes. Specifically, it proposes to use algorithms based on population analysis (relevant for large numbers of distributed nodes) to implement distributed training of deep neural networks.

In reviews and discussions, the reviewers and AC note missing or inadequate comparisons to previous work on asynchronous SGD, and possible lack of novelty compared to previous work. The reviewers also mentioned the incomplete empirical comparison to closely related work. On the writing, reviewers mentioned that the conciseness of the manuscript could be improved.